# Explore In-Context Message Passing Operator for Graph Neural Networks in A Mean Field Game

**Tingting Dan**
Department of Psychiatry
University of North Carolina at Chapel Hill
Chapel Hill, NC 27599, USA
Tingting_Dan@med.unc.edu

**Xinwei Huang**
Shanghai Fourth People's Hospital
School of Medicine, Tongji University
Shanghai, 200434, China
huanggenetics@tongji.edu.cn

**Won Hwa Kim**
Computer Science and Engineering
POSTECH
Pohang, Korea 37673, South Korea
wonhwa@postech.ac.kr

**Guorong Wu**[*]
Departments of Psychiatry and Computer Science
University of North Carolina at Chapel Hill
Chapel Hill, NC 27599, USA
grwu@med.unc.edu

## Abstract

In typical graph neural networks (GNNs), feature representation learning naturally evolves through iteratively updating node features and exchanging information based on graph topology. In this context, we conceptualize that the learning process in GNNs is a mean-field game (MFG), where each graph node is an agent, interacting with its topologically connected neighbors. However, current GNNs often employ the identical MFG strategy across different graph datasets, regardless of whether the graph exhibits homophilic or heterophilic characteristics. To address this challenge, we propose to formulate the learning mechanism into a variational framework of the MFG inverse problem, introducing an in-context selective message passing paradigm for each agent, which promotes the best overall outcome for the graph. Specifically, we seek for the application-adaptive transportation function (controlling information exchange throughout the graph) and reaction function (controlling feature representation learning on each agent), *on the fly*, which allows us to uncover the most suitable selective mechanism of message passing by solving an MFG variational problem through the lens of Hamiltonian flows. Taken together, our variational framework unifies existing GNN models into various mean-field games with distinct equilibrium states, each characterized by the learned in-context message passing operators. Furthermore, we present an agnostic end-to-end deep model, coined *Game-of-GNN*, to jointly identify the message passing mechanism and fine-tune the GNN hyper-parameters on top of the elucidated message passing operators. *Game-of-GNN* has achieved SOTA performance on diverse graph data, including popular benchmark datasets and human connectomes. More importantly, the mathematical insight of MFG framework provides a new window to understand the foundational principles of graph learning as an interactive dynamical system, which allows us to reshape the idea of designing next-generation GNN models.

---

[*]Corresponding author.

39th Conference on Neural Information Processing Systems (NeurIPS 2025).

# 1 Introduction

In a world of complex systems, graphs provide a powerful way to model relationships between objects [41]. Graph neural networks (GNNs) leverage graph structures to achieve success in diverse fields, such as social network analysis [13], biochemical engineering [15], and drug repurposing [11]. Despite their variations, most GNNs share key components: (1) feature representation for nodes and (2) message passing to propagate features across the graph [38]. Relational inductive biases, like permutation invariance, are also critical for aligning GNN designs with graph topology [5].

However, real-world graphs pose several challenges. The first lies in the heterogeneous relationship between graph topology and node labels. On heterophilic graphs, where connected nodes often differ in features, even simple multilayer perceptrons (MLPs) can outperform GNNs [45; 22]. To address this, researchers have explored adaptive message passing [24] and heterophily-aware measures [44]. Another key challenge is over-smoothing, where excessive aggregation blurs node distinctions. Solutions include diffusion-based models such as GRAND [7] and PDE-inspired approaches [36; 32]. Architectural innovations like residual networks [20] and Transformer backbones [16] have also been introduced to capture global dependencies and alleviate over-smoothing. Despite these advances, most GNN designs still rely heavily on domain-specific heuristics and backpropagation for tuning deep architectures, which remain fixed across all graph types. As illustrated by the gear-matching analogy in Fig. 1, such hand-crafted designs guided by domain expertise in machine learning and graph signal processing [27; 14], lack a system-level understanding of the learning mechanism. Consequently, achieving state-of-the-art accuracy does not guarantee that the model's inference principle is well aligned with the underlying graph structure. For instance, a particular GNN may perform well on homophilic graphs but poorly on heterophilic ones, or vice versa. To overcome these limitations, it is crucial to develop an explainable and principled framework for GNNs that provides mathematical guarantees and adaptability across diverse graph domains.

**GNN is a mean-field game.** To establish a principled learning framework for designing novel GNN architectures that generalize to unseen graphs, we conceptualize the learning process in GNNs as a *mean-field game* (MFG) involving a large number of interacting agents (i.e., graph nodes). In this view, each node acts as an individual player that optimizes its own feature representation while responding to the aggregated behavior of its neighbors, the so-called "mean field." Over successive message-passing iterations, all nodes collectively adjust their representations until reaching a stable equilibrium, analogous to players in a large-scale game who iteratively

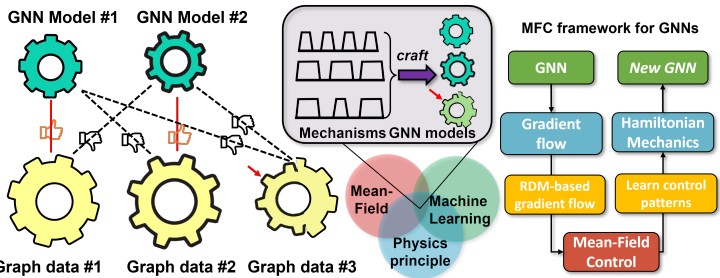

Figure 1: The motivation of our work. GNN instances and graph data intended for application are visually represented by gears of two different colors (green for models and yellow for data). The tooth pattern of data gears reflects the topological characteristics of graph data, such as whether it exhibits homophilic or heterophilic behavior. *Left*: Current approaches use the same GNN gear to match all data gears, relying on the "black-box" of gradient descent to fine-tune the underlying green gear (GNN model) to fit the yellow gears (data). This often leads to sub-optimal learning outcomes. *Right*: Our *Game-of-GNN* formulates graph learning as a mean-field game in the variational framework. By formulating the graph feature representation learning as a dynamical process, we capitalize on the equilibrium state (critical point of the mean-field control (MFC) problem) to form a Hamiltonian flow which allows us to link the "on-the-fly" design of GNN mechanism (shown in the blue box) and the discrete GNN instance for the real-world graph data. The showcase of our *Game-of-GNN* approach is the craft of suitable GNN mechanism (indicated by red arrows) for unseen graph data #3.

best-respond to the average strategy of the population. A central element of the MFG formulation is the temporal evolution of each agent's state. By interpreting the feature vector of each node as a form of potential energy, we further hypothesize that the underlying dynamics of GNNs follow the *second law of thermodynamics* [26], where the system evolves toward configurations that dissipate the negative entropy functional most efficiently. This perspective is supported by recent studies that interpret representation learning on graphs as a dynamic process of heat diffusion [7; 8], providing a unified physical and mathematical foundation for understanding GNN behavior.

**Reverse engineering GNN as a MFG inverse problem.** Following this notion, we propose to characterize the collective behaviors of simultaneous information exchange and update in GNNs as a mean-field game, where the dynamics of MFG is shaped by the second law of thermodynamics. In this context, the most promising learning mechanism for the specific graph data is characterized by an optimal transport from the initial feature representations to a Nash-equilibrium state (depending on the downstream learning tasks), where each graph node finds the best feature presentation for itself and the entire graph. Inspired by the mean-field theory [19], we further conceptualize that the dynamic learning process in GNNs forms a gradient flow that can be controlled by a set of learnable 'free energy functionals' derived from the underlying graph data. Thus, the search for the most appropriate functionals boils down to an MFG inverse problem. From the perspective of model explainability, the learned 'free energy functionals' act as *in-context message passing operators*, which allows us to explore novel learning mechanisms for the regime of graph data learning.

**Approach.** Taken together, we present a variational framework of MFG inverse problem to achieve the physics-informed learning paradigm of designing novel GNN mechanisms, where the optimal in-context message passing operators are essentially the mean-field control patterns associated with underlying graph data. The detailed explanation of our approach is shown in *Appendix A.4*. Specifically, the message passing operator consists of two functions of kinetic dynamics: (1) a function of transportation mobility for controlling node-to-node information exchange and (2) a function of reaction mobility for message update on each node. These two mobility functions determine the characteristic learning mechanism of information exchange and update within the specific GNN instance, by shaping the kinetic dynamics of the underlying 'free energy functional'. Since different graph data forms a unique dynamical system with distinct in-context message passing operators, our proposed physics-informed learning framework for GNN, coined as *Game-of-GNN*, integrates two hierarchical machine learning modules at both the mechanism and model instance levels. At the mechanism level, we seek the most suitable 'free-energy functionals' that shape the gradient flow in the MFG. Furthermore, we derive Hamiltonian flows as the governing equations of the underlying MFG problem. As multiple lines of work have demonstrated that GNN is equivalent to an underlying PDE [42; 8; 7; 36], the Hamiltonian flow becomes a stepping stone, which allows us to link the high-level learning mechanism in the variational framework of MFG and the fine-tuning hyperparameters at the GNN model instance level. The outcome of our work is an end-to-end deep model that jointly identifies the most suitable message passing operators and refines hyperparameters for the corresponding GNN instance.

The major technical contributions are four-fold. (1) We present a physics-informed learning framework for GNN that crafts the most suitable GNN model while performing machine learning on graph data. (2) We integrate the theory of mean-field game into graph neural networks which not only offers an in-depth understanding of GNNs but also provides a general guideline for developing deep models for unseen graph data. (3) We present a practical end-to-end solution, based on Hamiltonian flows, to customize the best GNN model for the underlying graph data. (4) In addition to the comprehensive evaluation on graph benchmark datasets, we explore the foundational principles of graph learning as an interactive dynamical system, which is valuable for the conceptual framework of developing future GNN models.

## 2 Methods

### 2.1 Background on Mean-Field Game

#### 2.1.1 Gradient flow for optimizing mean-field game

Suppose we have an undirected, weighted graph $\mathcal{G} = (\mathcal{V}, \mathcal{P})$ with $\mathcal{V} = \{v_i | i = 1, ..., N\}$ is a finite set of $N$ vertices and $\mathcal{P} \subset \mathcal{V} \times \mathcal{V}$ denotes the set of edges. The adjacency matrix is denoted as $A = [a_{ij}]_{i,j=1}^{N}$, where $[i, j] \in \mathcal{P}$. Suppose $x_t = \{x(v, t) | v \in \mathcal{V}\} \in \mathbb{R}^{N \times d}$ represent the distribution of graph feature embeddings (aka. potential energy) associated at time $t$. The continuity equation describes the evolution of the distribution $x_t$ can be formulated as $\frac{\partial}{\partial t} x_t = -div(x_t \gamma_t)$, where $div(\cdot)$ denotes the divergence operator and $\gamma_t$ is the latent velocity field. By constraining the evolution of $x_t$ being the gradient flow that minimizes the energy functional $\mathcal{E}(x_t) = \int_{\mathcal{G}} G(x_t) dv$, there exists a unique potential function $\Phi_t$ such that $\gamma_t = \nabla \Phi_t$ [21].

**Remark 1.** We remark that if $G(x(v,t)) = x(v,t) \log x(v,t)$, then the gradient flow $\frac{\partial x}{\partial t} = -div(x\nabla\Phi)$ satisfies the heat equation $\frac{\partial x}{\partial t} = -\Delta x$, where $\Phi(v,t) = G'(v,t) = \log x(v,t) + 1$. Mounting evidence shows that the message-passing mechanism $\frac{\partial x}{\partial t} = -\Delta x$ in graph convolutional networks can be formulated as a neural graph diffusion process [7].

### 2.1.2 Variational framework of MFG constrained by reaction diffusion model

Suppose $\Psi(v)$ is situated in the continuous domain of a smooth positive density space. First, we construct a Lyapunov functional $\mathcal{E}(\Psi(v)) = \int_{\mathcal{G}} G(\Psi(v))dv$, where $G : \mathbb{R} \to \mathbb{R}$ is convex with $G''(\Psi) > 0$. If the gradient flow satisfies the reaction diffusion model (RDM) $\frac{\partial \Psi}{\partial t} = \Delta F(\Psi) + R(\Psi)$, minimizing $\mathcal{E}(\Psi(v))$ forms a dynamical system:

$$\frac{\partial}{\partial t}\mathcal{E}(\Psi_t) = -g(\Psi_t)^{-1}\frac{\partial}{\partial \Psi}\mathcal{E}(\Psi_t), \quad g(\Psi) = \left(-\operatorname{div}\left(\frac{F'(\Psi)}{G''(\Psi)}\nabla\right) - \frac{R'(\Psi)}{G'(\Psi)}\right)^{-1}, \quad (1)$$

where $g(\Psi)$ is the weighted elliptic operator (*Proof in Appendix Sec. A.2.*). To foreshadow the motivation for introducing the notion of control pattern in the MFG framework (in Section 2.3.2), we define the following two functionals to simplify the analytic expression of $g(\Psi)$.

**Definition 1. Free-energy functional for transportation and reaction.** Given $F$, $R$, and $G$, we define the transportation mobility function $\Theta_1(\Psi) = \frac{F'(\Psi)}{G''(\Psi)}$ and reaction mobility function $\Theta_2(\Psi) = -\frac{R(\Psi)}{G'(\Psi)}$.

The derivation of $g(\Psi)$ not only links objective functional $\mathcal{E}(\Psi)$ and RDM-based gradient flow, but also allows us to define the mean-field information metrics [3] as follows.

**Definition 2. Mean-field information metric.** Denote $\sigma$ as a smooth, positive density function. Given the elliptic operator $g$ (in Eq. 1), the metric between two densities $\sigma_1$ and $\sigma_2$ is:

$$\zeta(\sigma_1, \sigma_2) = \int_{\mathcal{G}}(\nabla\sigma_1, \nabla\sigma_2)\Theta_1 dv + \int_{\mathcal{G}}(\sigma_1, \sigma_2)\Theta_2 dv \quad (2)$$

**Remark 2.** In the special case of $\Theta_1 = \Psi$ and $\Theta_2 = 0$ (first term in Eq. 2), the variational MFG problem seeks the optimal transport $\psi_1$ to move the mass from $\Psi_0$ to $\Psi_1$ by minimizing $L_2$-*Wasserstein* metric. In another special case that $\Theta_1 = 0$ and $\Theta_2 = \Psi$ (second term in Eq. 3), the variational problem is corresponding to the Fisher-Rao metric, which has been well studied in information geometry [2]. It is clear there are different choices of operator $g$ lead to different mean-field information metrics.

**Definition 3. Variational framework of mean-field game.** Given $\Theta_1$ and $\Theta_2$, a variational problem

$$\inf_{\psi_1, \psi_2, \Psi} \int_0^1 \left[\int_{\mathcal{G}} \frac{1}{2}\|\psi_1(v,t)\|^2\,\Theta_1(\Psi(v,t)) + \frac{1}{2}|\psi_2(v,t)|^2\,\Theta_2(\Psi(v,t))\,dv\right]dt \quad (3)$$

where the infimum is taken among all density functions $\Psi(v)$, vector fields $\psi_1$, and reaction rate functions $\psi_2$, such that

$$\partial_t\Psi(v,t) + \nabla\cdot(\Theta_1(\Psi(v,t))\psi_1(v,t)) = \psi_2(v,t)\Theta_2(\Psi(v,t)), \quad (4)$$

with fixed initial and terminal density functions $\Psi_0, \Psi_1$.

**Remark 3.** The example in **Remark 1** indicates that the dynamic process of graph learning can be framed as a variational problem governed by a predefined gradient flow. Furthermore, it is possible to identify the most appropriate combination of energy function $\mathcal{E}(\Psi)$ (Eq. 3) and gradient flow $\frac{\partial \Psi}{\partial t}$ (Eq. 4) using machine learning technique, which sets the stage for a 'meta-learning' paradigm for unseen graph data. In what follows, we first unify existing GNN models into a reaction-diffusion model (RDM): $\frac{\partial \Psi}{\partial t} = \Delta F(\Psi) + R(\Psi)$, where $F(\cdot)$ and $R(\cdot)$ are diffusion and reaction functions, respectively. By constraining the gradient flow to follow the characteristics of RDM, we further formulate the design of GNN instance into a variational framework of MFG inverse problem, seeking to uncover the 'free-energy functionals' (determining the kinetics of information exchange and update) that yield the best graph learning outcome in downstream applications. Together, we present a physics-informed approach to jointly perform machine learning using the most suitable message passing operators and fine-tune the GNN model instance through the Hamiltonian flows derived from the associated RDM.

## 2.2 Unifying GNNs into the Dynamics Shaped by Reaction Diffusion Model

In this section, we briefly review some representative GNNs and unify them in the umbrella of RDM. More details are shown in the Appendix A.

**GRAND** [7]. Graph neural diffusion (GRAND) draws inspiration from the heat diffusion equation, offering a unified mathematical framework for vanilla message-passing laws on graphs by $\frac{\partial \Psi}{\partial t} = \text{div}[c\nabla\Psi]$, where $F(\Psi) = \Psi$ and $R(\Psi) = 0$. To simplify the problem formulation here, we assume $c$ is a homogeneous and time-invariant diffusion function. Thus, the intrinsic diffusion-reaction pattern can be further simplified as $\frac{\partial \Psi}{\partial t} = \Delta\Psi$ after dropping $c$.

**GraphBel** [31]. Extended from GRAND, the Beltrami diffusion on graph (GraphBel) proposed to use Beltrami flow to normalize the graph gradient as $\frac{\partial \Psi}{\partial t} = \frac{1}{\|\nabla\Psi\|}\text{div}\left(\frac{\nabla\Psi}{\|\nabla\Psi\|}\right)$, where $\frac{\nabla\Psi}{\|\nabla\Psi\|}$ is a discrete analogue of the mean curvature operator. Without changing the diffusion-reaction property, we use $\Delta$ to indicate the normalized graph Laplacian operator here. Thus, the evolution of $\Psi$ becomes $\frac{\partial \Psi}{\partial t} = \|\nabla\Psi\|^{-1}\Delta\Psi$, where $\|\nabla\Psi\| = \langle \nabla\Psi, \nabla\Psi \rangle^{\frac{1}{2}}$ is time-invariant magnitude of graph gradient. Since $\frac{1}{\|\nabla\Psi\|}$ is decoupled with the divergence operator, it is straightforward to derive $F(\Psi) = \Psi$ and $R(\Psi) = 0$ in GraphBel.

**ACMP** [36]. Inspired by interacting particle dynamics, Allen-Cahn message-passing (ACMP) graph neural network models both attractive and repulsive forces between two connected nodes during message-passing process using a predefined Allen-Cahn double-well potential function $f(\Psi)$ [1]. The ACMP-based GNN models can be formulated as $\frac{\partial \Psi}{\partial t} = m_\Psi\left[\text{div}\left(\varepsilon_\Psi^2 \nabla\Psi\right) - f'(\Psi)\right]$, where $m_\Psi$ is a hyper-parameter for the mobility and $\varepsilon_\Psi$ a constant. Since $m_\Psi$ and $\varepsilon_\Psi$ do not determine the reaction-diffusion property, the PDE in ACMP can be simplified to $\frac{\partial \Psi}{\partial t} = \Delta\Psi - f'(\Psi)$. In this scenario, $F(\Psi) = \Psi$ and $R(\Psi) = -f'(\Psi)$.

It is apparent that the characteristic of information exchange and update in a particular GNN instance is determined by a gradient flow associated with the RDM. As we conceptualize that GNN is a mean-field game, gradient flow $\frac{\partial \Psi}{\partial t} = \Delta F(\Psi) + R(\Psi)$ eventually leads to the minimization of an energy functional $\mathcal{E}(\Psi) = \int_\mathcal{G} G(\Psi)$, which describes the collective behavior of agents in the MFG.

**Remark 4.** Current GNN models apply the same energy functional $\mathcal{E}(\Psi)$ across all graph data, regardless of whether the graph exhibits homophilic or heterophilic properties. It is analogous to using an identical strategy in a variety of games, irrespective of their differing rules. However, the gradient flow exhibits distinct dynamics depending on the choice of energy functional. In light of this, it is appealing to use the most appropriate energy functional $\mathcal{E}(\Psi)$ that promotes the highest reward in each mean-field game of GNN. Building on this idea, we seek to reshape the kinetic dynamics by introducing control patterns, yielding the in-context message passing operator.

## 2.3 A Variational Framework of MFG Inverse Problem for Designing Novel GNN Models

Following the spirit of mean-field theory, we frame the dynamic process of message passing in GNNs as a mean-field game, where optimal features emerge as the system reaches Nash equilibrium governed by the control patterns learned from the underlying graph data.

### 2.3.1 Graph neural network is a mean-field game

We set up a mean-field game with $N$ players in a continuum of non-cooperative rational agents (graph nodes) distributed spatially in the graph $\mathcal{G}$ and temporally in $[0,1]$. For an agent $v$ starting at $\Psi_0(v)$, the evolution of $\Psi(t,v)$ is completely determined by Eq. 4. To play the game over a time interval $[0,1]$, each agent seeks to minimize the objective functional in Eq. 3, where the transportation cost $\Theta_1$ and reaction mobility cost $\Theta_2$ are incurred by each agent's own action. In addition, the Hamilton–Jacobi–Bellman (HJB) equation equivalent to the variational formulation in **Definition 3** is given below.

**Proposition 1. HJB equation for mean-field game**. Assume $\Psi(v,t) > 0$ for $t \in [0,1]$. Then there exists a function $\Phi : \mathcal{G} \times [0,1] \to \mathbb{R}$, such that the critical points of variational problem Eq. 3 satisfy

$$\psi_1(v,t) = \nabla\Phi(v,t), \quad \psi_2(v,t) = \Phi(v,t) \tag{5}$$

with

$$\begin{cases} \partial_t \Psi(v,t) + \nabla \cdot (\Theta_1(\Psi(v,t))\nabla\Phi(v,t)) = \Theta_2(\Psi(v,t))\Phi(v,t), \\ \partial_t \Phi(v,t) + \frac{1}{2}\|\nabla\Phi(v,t)\|^2 \Theta_1'(\Psi(v,t)) + \frac{1}{2}|\Phi(v,t)|^2 \Theta_2'(\Psi(v,t)) = 0, \end{cases} \tag{6}$$

and $\Psi(v, 0) = \Psi_0(v), \quad \Psi(v, t) = \Psi_1(v)$.

*Sketch of proof.* We introduce $\Phi$ as the Lagrange multiplier of variational problem (Eq. 3) constrained by the gradient flow in Eq. 4. Then we derive the solution of vector field $\psi_1$, reaction function $\psi_2$, and Hamiltonian flow in Eq. 6 by following the schema of saddle point problem. The detailed proof is shown in *Appendix Sec. A.3*.

**Remark 5.** If $\Theta_1 = \Psi$ and $\Theta_2 = 0$, the above formulation corresponds to the well-known Benamou-Brenier formula [6] in optimal transport. If $\Theta_1$ and $\Theta_2$ are positive functions then the objective functional in Eq. 3 is convex, making the derived gradient flow in Eq. 6 a minimizer of the variational MFC problem [21].

**Remark 6.** Suppose $\Psi_0$ is the initial graph representations. Given $\Theta_1$ and $\Theta_2$, Proposition 1 indicates that, at the mechanism level, the dynamical mechanics of feature representation learning from $\Psi_0$ to $\Psi_1$ is characterized by a Hamiltonian flow (Eq. 6), while at the model instance level, the alignment between the learned features $\Psi_1$ (terminal state) and the downstream task can be fine-tuned using a PDE-based GNN approach [42] which is governed by the Hamiltonian flow.

Given $\Psi_0$ and $\Psi_1$, a natural question is: *What is the most efficient way to transport $\Psi_0$ to $\Psi_1$?* The key to answering this optimal transport question is to study the critical point of the objective functional $\mathcal{E}(\Psi)$ with respect to $\Theta_1$ and $\Theta_2$, which leads to the MFG inverse problem.

### 2.3.2 Discover in-context message passing operators through MFG inverse problem

The variational MFG framework provides a potential optimal solution for the objective functional in Eq. 3 by examining the saddle point. Recall GNN is a mean-field game. The physical principle is characterized by the pre-selected transportation functional $\Theta_1$ and reaction functional $\Theta_2$. In contrast to the special cases of 2-Wasserstein distance (where $\Theta_1(\Psi) = \Psi, \Theta_2(\Psi) = 0$) and Fish-Rao metric (where $\Theta_1(\Psi) = 0, \Theta_2(\Psi) = \Psi$), $\Theta_1$ and $\Theta_2$ are essentially the weighted functions on each location $v$, acting as the expected **in-context message passing operator** that allow us to regulate the local message exchange and update during the evolution of graph representations $\Psi_t$. Naturally, we are motivated to learn the in-context message passing operators $(\Theta_1, \Theta_2)$, from the underlying graph data to improve the performance of GNN models.

In light of this, we present the following meta-learning paradigm that derives the most suitable learning mechanism from MFG inverse problem and meanwhile optimizes model parameters using GNN backbones. By doing so, we expect to (1) enhance graph data learning performance on top of the existing GNN models and (2) establish an in-depth understanding of how individual node learns the best feature representations for themselves and the entire graph. Specifically, we introduce the functional Hamilton-Jacobi equations in positive density space (i.e., graph space) and define a Hamilton functional $\mathcal{H} : \mathcal{G} \times \mathcal{G} \rightarrow \mathbb{R}$ as follows:

$$\mathcal{H}(\Psi, \Phi) = \int_{\mathcal{G}} \left( \frac{1}{2} \|\nabla\Phi\|^2 \Theta_1(\Psi) + \frac{1}{2} |\Phi|^2 \Theta_2(\Psi) \right) dv, \tag{7}$$

where the density function $\Psi$ serves as the state variable (akin to position), while the potential function $\Phi$ acts as the momentum variable in graph space.

### 2.4 Reverse Engineering GNNs by In-context Message Passing Operators

In Sec. 2.2, we have shown the relationship between GNN model instance and reaction-diffusion equation. Despite many GNN models being fundamentally linked to the same PDE, they exhibit varied learning behaviors, yielding distinct learned feature representations. Within the variational framework of the MFG inverse problem, such diversity can be attributed to the fact that different GNNs are driven by distinct objective functionals $\mathcal{E}(\Psi)$, each governed by unique physical principles. In Table 1, we summarize the energy variational functional $\mathcal{E}$, mobility functions $\Theta_1$ and $\Theta_2$, reaction-diffusion equation, and the corresponding Hamiltonian flows.

It is clear that the objective functional $\mathcal{E}(\Psi) = \int_{\mathcal{G}} G(\Psi) dv$ (for crafting GNN mechanism) and the associated gradient flow in $\frac{\partial \Psi}{\partial t} = \Delta F(\Psi) + R(\Psi)$ (for optimizing GNN instance) are both related to transportation mobility function $\Theta_1$ and reaction mobility function $\Theta_2$. By capitalizing on this property, our *Game-of-GNN* emerges as a meta-learning graph learning approach. For clarity, we summarize how GNN is formulated as mean-field games in *Appendix A.4*.

Table 1: Variational functionals $\mathcal{E}(\Psi) = \int_{\mathcal{G}} G(\Psi)dv$, diffusion function $F(\cdot)$, reaction function $R(\cdot)$, mobility functions $\Theta_1(\cdot)$ and $\Theta_2(\cdot)$, and Hamiltonian equations.

| Model | $\mathcal{E}(\Psi) = \int_{\mathcal{G}} G(\Psi)dv$ | $F(\Psi)$ | $R(\Psi)$ | $\Theta_1(\Psi) = \frac{F'}{G''}$ | $\Theta_2(\Psi) = -\frac{R}{G'}$ | Hamiltonian Equation |
|---|---|---|---|---|---|---|
| **GRAND** | $\int_{\mathcal{G}}(\Psi\log\Psi - 1)dv$ | $\Psi$ | $0$ | $\Psi$ | $0$ | $\begin{cases} \partial_t\Psi + \nabla\cdot(\Psi\nabla\Phi) = 0 \\ \partial_t\Phi + \frac{1}{2}\|\nabla\Phi\|^2 = 0 \end{cases}$ |
| **GraphBel** | $\int_{\mathcal{G}}(\frac{1}{2}\Psi^2)dv$ | $\Psi$ | $0$ | $1$ | $0$ | $\begin{cases} \partial_t\Psi + \nabla\cdot(\nabla\Phi) = 0 \\ \partial_t\Phi = 0 \end{cases}$ |
| **ACMP** | $\int_{\mathcal{G}} f(\Psi)dv$ | $\Psi$ | $-f'(\Psi)$ | $f''(\Psi)^{-1}$ | $1$ | $\begin{cases} \partial_t\Psi + \nabla\cdot(f''(\Psi)^{-1}\nabla\Phi) - \Phi = 0 \\ \partial_t\Phi - \frac{1}{2}\|\nabla\Phi\|^2\frac{f'''(\Psi)}{f''(\Psi)^2} = 0 \end{cases}$ |

**Network architecture for *Game-of-GNN*.** Inspired by [42], we propose an agnostic end-to-end deep model based on Hamiltonian mechanics, which characterizes information propagation in graph networks using a Hamiltonian-like structure. The implemented details are shown in Algorithm 1. Specifically, we regard the potential energy $\Psi$ and latent function $\Phi$ ($\nabla\Phi$ is a flow vector field) in Eq. 6 as the position and momentum variables, respectively, in the Hamiltonian system, where the phase space $(\Psi, \Phi)$ characterizes the system's evolution. Prior to $(\Psi(0), \Phi(0))$, we deploy a set of fully-connected layers $\mathcal{F}$ to project the observed nodal features $x$ to the energy function. There are two major inter-connected network components in *Game-of-GNN*: (1) meta-learning component $\mathcal{O}$ for generating message-passing operators $\Theta_1$ and $\Theta_2$ based on the current estimation of phase space $(\Psi, \Phi)$; and (2) PDE-based GNN instance $\mathcal{M}$ for solving the evolution of Hamiltonian flow, where the terminal state of Hamiltonian flow is used to plug-in with the down-stream learning task. The connection between $\mathcal{M}$ and $\mathcal{H}$ is the learned message passing operators $\Theta_1$ and $\Theta_2$.

---

**Algorithm 1:** *Game-of-GNN* algorithm

---

**Input:** Graph $\mathcal{G} = (\mathcal{V}, \mathcal{P})$, node features $x(t)$, adjacency matrix $A$
**Output:** The mobilities $\Theta_1, \Theta_2$, the evolved node feature representation $x(T)$

1 **for** $i = 1 \ldots |\mathcal{V}|$ **do**
2      Construct phase space by $(\Psi_i, \Phi_i) \leftarrow \mathcal{F}(x_i(t))$;
3      **for** $t = 1 \ldots T$ **do**
4          Learn mobilities $\Theta_1, \Theta_2$ by $\Theta_1, \Theta_2 \leftarrow \mathcal{O}((\Psi, \Phi))$;
5          Construct Hamiltonian function $\mathcal{M}(\Psi_i(t), \Phi_i(t))$ by Eq. 7;
6          Build PDE of the evolution of system state on graph by Eq. 9;
7          Derive the trajectory $(\Psi_i(t), \Phi_i(t))$ by PDE solver;
8      **end**
9      Yield the evolved node feature representation $x(T)$ by $x(T) \leftarrow \Pi(\Psi(T), \Phi(T))$;
10 **end**

---

$\mathcal{O}$: **Generating $\Theta_1$ and $\Theta_2$ by input convex neural network (ICNN).** Since energy function $\mathcal{H}(\Psi, \Phi)$ is completely determined by mobility function $\Theta_1$ and $\Theta_2$ (shown in Eq. 7), we propose to use a neural network $\mathcal{O}$ to establish the implicit mapping between the input $(\Psi, \Phi)$ and output $(\Theta_1, \Theta_2)$. As a crucial prerequisite for deriving the Hamiltonian flow outlined in Eq. 6, the objective function needs to be convex. Therefore, we use input convex neural network [4] as the backbone of $\mathcal{O}$, which yields the convex function instance in a recursive manner:

$$z^{(k+1)} = \sigma^{(k)}\left(W_1^{(k)}z^{(k)} + W_2^{(k)}(\Psi, \Phi) + b^{(k)}\right), \tag{8}$$

where $z^{(k)}$ denotes the output of $k^{th}$ layer. Each layer consists of two MLPs which project (1) the output from the previous layer $z^{k-1}$ (parameterized by $W_1^{(k)}$) and (2) the current phase-space $(\Psi, \Phi)$ (parameterized by $W_2^{(k)}$) and concatenate the output of two MLPs into $z^{(k+1)}$ by applying a non-linear activation $\sigma^{(k)}$ with a bias vector $b^{(k)}$. Thus, the output of meta-learning component $\mathcal{O}$ is the transportation function $\Theta_1$ and reaction function $\Theta_2$, which allows to define the message passing operator for each graph node based on the phase space $(\Psi, \Phi)$.

$\mathcal{M}$: **GNN model based on Hamiltonian flow.** In physics, systems evolve according to fundamental physical laws, with a (pre-defined) conserved quantity function $\mathcal{H}(\Psi, \Phi)$ that remains constant along the system's trajectory of evolution. This conserved quantity is commonly interpreted as the 'system

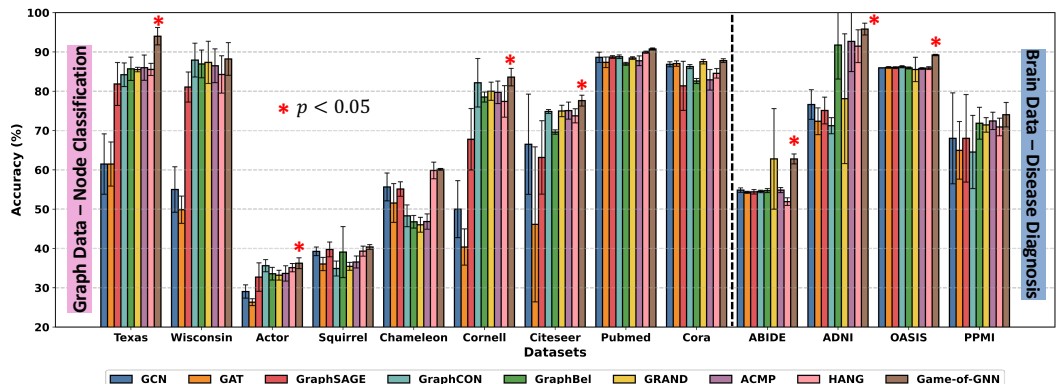

Figure 2: *Left*: Test accuracies (%) on nine graph networks for node classification task. Statistical significance is assessed based on 20 resampling tests conducted using a randomized seed. '∗' means statistically significance with $p \leq 0.05$. *Right*: Diagnosis accuracies (%) on disease-based datasets.

energy'. We model the evolution of graph feature representations by following Hamiltonian equation:

$$\partial_t \Psi = \frac{\delta}{\delta \Phi} \mathcal{H}(\Psi, \Phi), \quad \partial_t \Phi = -\frac{\delta}{\delta \Psi} \mathcal{H}(\Psi, \Phi), \tag{9}$$

with the initial features $(\Psi(0), \Phi(0))$ at time $t = 0$ being the vectors of potential energies. Supposing $\Psi(v) \in \mathbb{R}^d$ and $\Phi(v) \in \mathbb{R}^d$, we generate a $2d$-dimensional vectors that are then split into two equal halves: the first half serves as the feature (position) vector $\Psi$, while the second half represents the momentum vector guiding the system's evolution. Assuming a terminal time point $t = T$, the solution of the system is represented by $\Psi(T)$ and $\Phi(T)$, obtained through integration to derive the trajectory $(\Psi(t), \Phi(t))$ described in Eq. 9. After that, we apply the canonical projection function $\Pi$ to extract the concatenated feature vector $\Psi$ of the nodes from $(\Psi(T), \Phi(T))$, yielding $\Pi(\Psi(T), \Phi(T)) \rightarrow x(T)$, which can then be utilized for downstream tasks such as node classification.

## 3   Experiments

**Experiment setup.** The evaluation on *Game-of-GNN* not only includes benchmark with respect to existing state-of-the-art GNN models but also a *proof-of-concept* exploration to uncover novel insights into graph learning. Specifically, benchmark tests include (1) node classification and (2) graph classification. For **graph node classification**, we apply our method to both heterophilic and homophilic datasets (sorted by homophily ratio $h$ [45]): Texas ($h = 0.11$), Wisconsin ($h = 0.21$), Actor ($h = 0.22$), Squirrel ($h = 0.22$), Chameleon ($h = 0.23$), Cornell ($h = 0.3$), Citeseer ($h = 0.74$), Pubmed ($h = 0.8$) and Cora ($h = 0.81$), where $h$ indicate the fraction of edges that connect nodes with the same label. For **graph classification**, we *first* conduct an experiment on the benchmark results on TUDataset [25] including MUTAG, NCI1, ENZYMES, D&D, PTC_FM, IMDB-B and PROTEINS. To demonstrate the generality and scalability of our proposed model, we *then* apply the *Game-of-GNN* to human connectomes for disease diagnosis, we use the processed neuroimaging data in the published datasets [39]: ABIDE (Autism), ADNI (Alzheimer) [37], OASIS (Alzheimer) [18], PPMI (Parkinson), where we use regional BOLD (blood oxygenation level-dependent) time series as the graph embedding and functional connectivity (FC) [9] with automated anatomical labeling (ALL) atlas (116 regions) [33] as the adjacency matrix. The data description is shown in *Sec. B.1*.

We compare the performance with various benchmark GNN models, including vanilla GCN [17], GAT [34], GraphSAGE [13], GraphCON [30], GraphBel [31], GRAND [7], ACMP [36] HANG [42], GIN [40] and AM-GCN [35]. For conventional graph data in node classification experiments, we follow a challenging data-splitting method published in [43] (graph robustness benchmark), with 60% for training, 10% for validation, and the rest of the nodes for the testing set. For TUDataset, we report the 10-fold cross-validation (follow [29]) results on different models. For human brain data, we report the 5-fold cross-validation results.

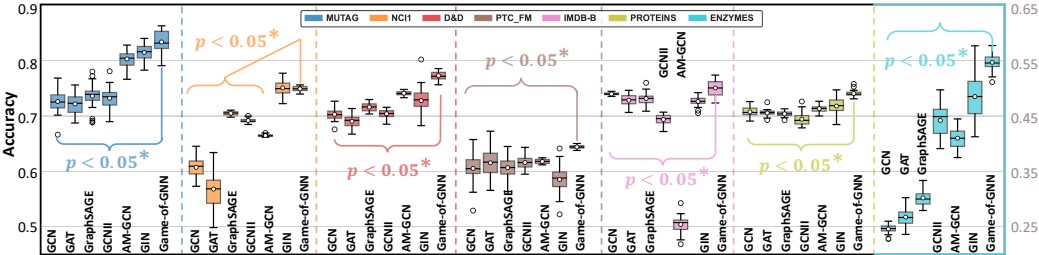

Figure 3: Performance on TUDataset across seven methods.

## 3.1 Benchmark Evaluations

**Performance on graph node classification.** Fig. 2 (left) lists the comparison results for nine classic graph datasets on nine methods. *Game-of-GNN* achieves SOTA performance on both heterophilic and homophilic over the existing hand-designed GNN models. Moreover, we perform an ablation study in terms of depth of ICNN, the result is shown in *Appendix C.* **Discussion.** These results provide evidence that our variational framework is able to customize the most suitable mobility functions for different graph data, which contributes to enhanced learning performance compared to other 'one-size-fits-all' approaches. This improvement primarily stems from the design of $\Theta_1$ and $\Theta_2$, which enable the model to effectively adapt to heterophilic structures. Specifically, $\Theta_1(\Psi)$ acts like a *feature-dependent attention weight*, where on heterophilic nodes, $\Psi$ shows high neighbor variance leading to $\Theta_1 \downarrow$ and hence *less smoothing*. In contrast, $\Theta_2(\Psi)$ behaves like a *feature-dependent residual gate*, where high neighbor variance results in $\Theta_2 \uparrow$, indicating *stronger self-correction*. By making both transport ($\Theta_1$) and reaction ($\Theta_2$) depend on local feature statistics $\Psi$, our framework *automatically throttles neighbor averaging and boosts self-repulsion* in heterophilic regions, providing a principled mechanism for handling heterophily.

**Performance on graph classification.** We include a 'global_max_pool' function and a fully connected layer to achieve the graph classification task. Fig. 3 presents the benchmark results for six classic methods and our *Game-of-GNN* on the popular TUDataset. Our *Game-of-GNN*, demonstrates decent performance across various types of graph data, including molecules, bioinformatics, and social networks, outperforming several existing hand-designed GNN models. **Discussion.** These results provide compelling evidence that our variational framework is well-suited for various types of graph datasets, resulting in improved learning performance compared to approaches tailored to specific datasets.

**Benchmark on human connectomes.** Fig. 2 (right) summarizes the diagnostic performance across four disease-based datasets, where we predict the risk of developing neurological disease in unseen subjects using graph data. The experimental findings demonstrate that our method exhibits significant effectiveness in disease diagnosis, suggesting the promising clinical value of deploying our approach in disease early diagnosis. **Discussion.** Along with atypical neuron growth/loss [23], many neurological diseases manifest network dysfunction syndromes [28]. In addition to the standard attention mechanism in GNN [34], the variational framework in *Game-of-GNN* allows us to uncover the dynamic mechanism of disease progress from a system perspective, as shown below.

## 3.2 Method Exploration: New Insight of Graph Learning Beyond Attention

In this section, we put the spotlight on the transport function $\Theta_1$ since this mobility function is intuitively relevant to the message-exchanging mechanism in GNNs. For each graph dataset, one of the outputs of *Game-of-GNN* is the learned $\Theta_1$ at each graph node, where we essentially employ ICNN backbone (Eq. 8 in the learning module $\mathcal{O}$) to generate a convex function based on the flow information $\Psi$. Assuming the latent convex function is a polynomial function, we compute the mean polynomial power $\alpha$ at each graph node by applying uni-variate polynomial fitting for each element and then averaging the degrees of polynomial power. After that, we conduct several post-hoc analyses at graph level and node level, respectively. *First*, we use the averaged polynomial power (across nodes) to express the graph homophily ratio $h$, to uncover the new insight into how the dynamics of information exchange in GNN correlates with the properties of the graph data. *Second*, we extend this global analysis to each graph node with the hypothesis that mobility of spreading node embeddings (related to neuropathology burdens) underlies the biological mechanism in disease progression.

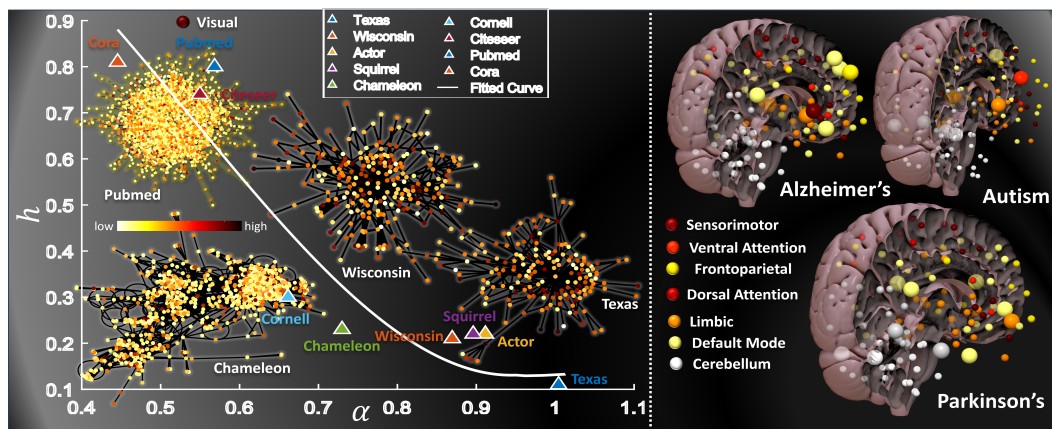

Figure 4: Left: Correlation between graph homophily ratio $h$ (y-axis) and the learned control pattern $\Theta_1(\Psi) = \Psi^\alpha$ (x-axis). $h$ is a global measurement where a large degree indicates a better alignment between graph topology and the consistency of labels across connected nodes. We use a polynomial function to fit $\Theta_1$ for each node, where the degree of $\alpha$ is inversely proportional to the freedom of local information exchange. It is clear that the learned control pattern is highly correlated with the heuristic measurement. Right: Top ten significant brain regions associated with the pathophysiological mechanism of AD, Autism, and PD.

*Results.* As the white curve shown in Fig. 4 left, the $\alpha \sim h$ relationship across nine graph dataset indicates a notable anti-correlation (more detailed analysis is provided in Appendix C). It implies that the effective way to perform graph learning is to promote information exchange on homophilic graphs (such as Cora and Pubmed) while constraining the diffusion of information between connected nodes with different labels in heterophilic graphs (such as Texas and Wisconsin). The reason behind is rooted in the MFG objective functional (Eq. 3 that larger degree of $\Theta_1$ encourages the optimization process favoring smaller flows $\Psi_1$ which is aligned with the heuristic of penalizing information exchange in heterophilic graphs. Furthermore, we display the learned $\Theta_1$ at node level for Texas, Wisconsin, Chameleon, and Pubmed in Fig. 4 left, where bright yellow and dark red denote for small and large degree of $\Theta_1(\Psi(v))$, respectively. Importantly, we find that *global homophily $h$ alone does not fully explain the performance differences across datasets*. As shown in Texas, Chameleon, and Pubmed (as shown in Fig. 2 left), datasets with similar $h$ can still exhibit markedly different gains due to variations in *local structure* and *feature quality*, further underscoring the importance of adapting mobility functions beyond global homophily.

*Discussion.* Similarly, we conduct the same post-hoc analysis to investigate the biological underpinning between the learned node-wise transport mobility degree and pathophysiological mechanism of disease progression. In Fig. 4 right, we use large size node to indicate the larger mobility of the underlying node (associated with smaller degree of $\Theta_1(\Psi(v))$). It is interesting to find that the brain regions with high dynamics for pathology propagation are closely associated with our current findings on disease etiology. Take Alzheimer's disease (AD) for example, resting-state fMRI studies have identified significant alterations in BOLD signal dynamics within the default mode network (DMN), which may indicate abnormalities in functional connectivity [12]. Here, we use machine learning techniques to provide another piece of data-driven evidence to support this finding as most of the large-size nodes are located in DMN. Additionally, our findings reveal that (1) increased mobility of pathological factors in the cerebellum correlates with the progression of Parkinson's disease, and (2) accelerated neuron overgrowth in the dorsal attention and limbic networks, as well as the cerebellum, may potentially be a contributing factor to autism. These promising results underscore the new window to answer neuroscience questions using explainable machine learning techniques.

## 4 Conclusions

In this work, we integrate the theory of mean-field game into GNNs to enhance our understanding and guide the development of deep models for new graph datasets. We also provide an end-to-end solution using Hamiltonian flows to jointly learn suitable message passing operators for GNN model and fit the customized GNN model to the underlying graph data. Our approach is thoroughly evaluated on standard benchmark datasets, and we explore fundamental principles of graph learning as an interactive dynamical system, which not only advances GNN understanding but also contributes to the broader field of graph-based machine learning.

## Acknowledgement

This work was supported by the National Institutes of Health (AG091653, AG068399, AG084375) and the Foundation of Hope. The views and conclusions contained in this document are those of the authors and should not be interpreted as representing the official policies, either expressed or implied, of the NIH.

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

# A    Proof of Propositions And Detailed Formulations And Explanations

## A.1    The Explanation of Definition 1

The motivation of formulating $\Theta_1 = \frac{F'(\Psi)}{G''(\Psi)}$ and $\Theta_2 = -\frac{R(\Psi)}{G'(\Psi)}$ is to facilitate (1) linking the diffusion term $F(\Psi)$ and reaction term $R(\Psi)$ in RDM to energy functional $\mathcal{E}(\Psi) = \int_{\mathcal{G}} G(\Psi(v)) dv$ and (2) generalizing existing GNNs into the RDM framework. As described in Sec. 2.2, current PDE-based GNN models can be regarded as the RDM with empirically defined diffusion functional instance $F(\Psi)$ and reaction functional instance $R(\Psi)$. In our approach, the behavior of diffusion $F$ and reaction $R$ is not fixed for all graph data. Instead, we learn the most appropriate diffusion and reaction functions through the transport functional $\Theta_1$ and reaction mobility functional $\Theta_2$.

## A.2    Proof of Eq. (1)

To prove Eq. (1), we consider constructing a Lyapunov functional $\mathcal{E} : \mathcal{G} \to \mathbb{R}$ to study the RDM, thus considering $\mathcal{E}(\Psi) = \int G(\Psi(v)) dv$, where $G : \mathbb{R} \to \mathbb{R}$ is a convex function with $G''(\Psi) > 0$. In such cases, we have

$$
\frac{d}{dt}\mathcal{E}(\Psi(t,\cdot)) = \int G'(\Psi(t,v)) \cdot \partial_t \Psi(t,v) dv = \int G'(\Psi(t,v))(\Delta F(\Psi(t,v)) + R(\Psi(t,v))) dv
$$

$$
= -\int (\nabla G'(\Psi(t,v)), \nabla G'(\Psi(t,v))) \frac{F'(u(t,v))}{G''(\Psi(t,v))} dv + \int G'(\Psi(t,v))^2 \frac{R(\Psi(t,v))}{G'(\Psi(t,v))} dv
$$

(10)

where we apply $\nabla G'(\Psi) = G''(\Psi)\nabla\Psi$ in Eq. 10. Herein, we assume $-\frac{R}{G'} > 0$ and $F'(\Psi) > 0$ for $\Psi > 0$, thus we have $\frac{d}{dt}\mathcal{E}(\Psi) \leq 0$, indicting that functional $\mathcal{E}(\Psi)$ is not increasing along flow. The decay behavior described above suggests a gradient flow formulation for the dynamics outlined in RDM. To refine and clarify this concept, we introduce an inverse of the weighted elliptic operator

$$
g(\Psi) := \left( -\nabla \cdot \left( \frac{F'(\Psi)}{G''(\Psi)} \nabla \right) - \frac{R(\Psi)}{G'(\Psi)} \right)^{-1}
$$

(11)

Thus we have

$$
\partial_t \Psi = -g(\Psi)^{-1} \frac{\delta}{\delta\Psi}\mathcal{G}(\Psi) = -\left( -\nabla \cdot \left( \frac{F'(\Psi)}{G''(\Psi)} \nabla \right) - \frac{R(\Psi)}{G'(\Psi)} \right) \frac{\delta}{\delta\Psi}\mathcal{G}(\Psi)
$$

$$
= \nabla \cdot \left( \frac{F'(\Psi)}{G''(\Psi)} \nabla G'(\Psi) \right) + \frac{R(\Psi)}{G'(\Psi)} G'(\Psi) = \Delta F(\Psi) + R(\Psi)
$$

(12)

where $\frac{\delta}{\delta\Psi}$ denotes the $L^2$ first variation w.r.t. $\Psi \in \mathcal{M}(\mathcal{E})$. Based on the above notation, the dissipation of Lyapunov functional $\mathcal{E}$ along RDM satisfies $\frac{d}{dt}\mathcal{E}(\Psi) = -\int \left( \frac{\delta}{\delta\Psi}\mathcal{G}(\Psi), g(\Psi)^{-1} \frac{\delta}{\delta\Psi}\mathcal{G}(\Psi) \right) dv \leq 0$.

## A.3    Proof of Proposition 1

To prove Proposition 1, we first rewrite the variables in variational problem Eq. 3 of the main text as

$$
q_1(t,v) = \Theta_1(\Psi)\psi_1(t,v), \quad q_2(t,v) = \Theta_2(\Psi)\psi_2(t,v),
$$

(13)

Thus the variational problem Eq. 3 forms

$$
\inf_{m_1,m_2,u} \left\{ \int_0^1 \int_{\mathcal{G}} \frac{\|q_1(t,v)\|^2}{2\Theta_1(\Psi(t,v))} + \frac{|q_2(t,v)|^2}{2\Theta_2(\Psi(t,v))} dvdt : \right.
$$

$$
\left. \partial_t \Psi(t,v) + \nabla \cdot q_1(t,v) = q_2(t,v), \quad \text{fixed } \Psi_0, \Psi_1 \right\}.
$$

(14)

Denote the Lagrange multiplier of Eq. 14 by $\Phi$. We consider the following saddle point problem

$$
\inf_{q_1,q_2,\Psi} \sup_{\Phi} \mathcal{L}(q_1, q_2, \Psi, \Phi),
$$

(15)

with

$$\mathcal{L}(q_1, q_2, \Psi, \Phi) = \int_0^1 \int_{\mathcal{G}} \left\{ \frac{\|q_1(t,v)\|^2}{2\Theta_1(\Psi(t,v))} + \frac{|q_2(t,v)|^2}{2\Theta_2(\Psi(t,v))} \right. \tag{16}$$
$$\left. + \Phi(t,v) \left( \partial_t \Psi(t,v) + \nabla \cdot q_1(t,v) - q_2(t,v) \right) \right\} dv dt.$$

By finding the saddle point of $\mathcal{L}$, we have

$$\begin{cases} \frac{\delta}{\delta q_1}\mathcal{L} = 0, \\ \frac{\delta}{\delta q_2}\mathcal{L} = 0, \\ \frac{\delta}{\delta \Psi}\mathcal{L} = 0, \\ \frac{\delta}{\delta \Phi}\mathcal{L} = 0, \end{cases} \Rightarrow \begin{cases} \frac{q_1}{\Theta_1} = \nabla\Phi, \\ \frac{q_2}{\Theta_2} = \Phi, \\ -\frac{1}{2}\frac{\|q_1\|^2}{\theta_1^2}\Theta_1' - \frac{1}{2}\frac{|q_2|^2}{\Theta_2^2}\Theta_2' - \partial_t\Phi = 0, \\ \partial_t\Psi + \nabla \cdot q_1 - q_2 = 0, \end{cases} \tag{17}$$

where $\frac{\delta}{\delta q_1}, \frac{\delta}{\delta q_2}, \frac{\delta}{\delta \Psi}, \frac{\delta}{\delta \Phi}$ are $L^2$ first variations w.r.t functions $q_1, q_2, \Psi, \Phi$, respectively. After that, by substituting the above two row equations into the last two row equations of Eq. 17, we derive the PDE pair Eq. 6 in the main text.

## A.4 GNN Is A Mean Filed Game

In the following, we emphasize the explanation of the principle of how GNN is formulated as a mean-field game.

GNN is a dynamical system. Simply put, GNN is a black box that converts the initial feature representations into a latent subspace by a set of information exchanges (constrained by graph topology) and projection (using a mapping function shared by all graph nodes). As GNNs often consist of multiple layers, the evolution of feature representation from the initial state (input graph embeddings) to the terminal state (last layer of GNN) can be regarded as a time-dependent dynamical system, where the dynamics is determined by a governing equation (in the form of PDE). In the reminiscent of the Brachistochrone problem [2] (a classic physics problem that involves finding the curve down which a bead sliding under the influence of gravity will travel in the least amount of time between two points), the powerful calculus of variations (COV) allows us to generate various governing equations, providing a necessary condition that a function must satisfy in order to be an extremum of a given functional.

GNN-PDE-COV interplay. Inspired by recent PDE-based deep models such as Neural ODE and GRAND, we frame the layer-by-layer feed-forward process $x^{(l+1)} = \sigma(AWx^{(l)})$ as a dynamical system, where the time-evolving mechanics is determined by the graph heat equation $\frac{\partial x(t)}{\partial t} = \Delta x(t)$. Here, $A$ and $\Delta$ denote the normalized adjacency matrix and Laplacian matrix, $W$ is the learnable mapping parameters, and $\sigma$ denotes the nonlinear activation function. Indeed, the evolution of the heat equation forms the gradient flow of the Dirichlet energy $\varepsilon(x) = \frac{1}{2}\int |\nabla x|^2$. Thus, we have established a connection between the GNN model instance in the discrete domain and the equivalent variation functional in the continuous domain, where the governing equation is acting as a stepping stone.

Setup of mean-field game for GNN. Following the notion of mean-field game, each graph node is acting as an agent. The game is to find the best feature presentations for all graph nodes that minimize the loss function in GNN. In a mean field game, each agent (aka. graph node) makes decisions based on both their individual state and the aggregate effect of the states and actions of all other agents, often referred to as the "mean field." The primary goal is to find a Nash equilibrium, which is a strategic decision-making in very large populations of interacting agents such that no agent can benefit by changing their strategy while others keep theirs unchanged. Mathematically, mean field games often involve solving coupled partial differential equations such as Hamilton-Jacobi-Bellman (HJB) equation which describes the optimal control problem for the evolution of the distribution of agents' states over time. In our work, we introduce mean-field game and mean-field control (MFC) into GNN, as described below.

A MFC framework for designing a novel GNN model. First, we extend the heat equation to a graph-based reaction-diffusion model, where the system behavior is determined by a mobility functional

---
[2]https://en.wikipedia.org/wiki/Brachistochrone_curve

$(\Theta_1)$ and reaction functional $(\Theta_2)$. Second, we follow the recent work of MFC [21] to define the main variational problem, consisting of a metric space (Eq. 3) and gradient flow (Eq. 4). After that, we study the critical point of the variational problem, yielding a Hamiltonian flow in MFC problem (Proposition 2).

Takeaway. We formulate the dynamic process of graph feature representation as a mean-field game where the game policy is defined in a mean-field control perspective. In the real application, our model simultaneously (1) crafts GNN model instance by identifying the most appropriate game policy (i.e., derive the instance of mobility functional $\Theta_1$ and reaction functional $\Theta_2$), and (2) optimizes GNN instance using Hamiltonian flow.

## B  Experimental Details

### B.1  Datasets and Hyperparameters

*Classic graph data for node classification.* We summarize the data information in the following Table 2.

Table 2: Data description for node classification.

|  | **Texas** | **Wisconsin** | **Actor** | **Squirrel** | **Chameleon** | **Cornell** | **Citeseer** | **Pubmed** | **Cora** |
|---|---|---|---|---|---|---|---|---|---|
| **Hom. ratio** $h$ | 0.11 | 0.21 | 0.22 | 0.22 | 0.23 | 0.3 | 0.57 | 0.74 | 0.81 |
| **#Nodes** $|\mathcal{V}|$ | 183 | 251 | 7,600 | 5,201 | 2,277 | 183 | 3,327 | 19,717 | 2,708 |
| **#Edges** $|\mathcal{P}|$ | 295 | 466 | 26,752 | 198,493 | 31,421 | 280 | 4,676 | 44,327 | 5,278 |
| **#Classes** $|\mathcal{Y}|$ | 5 | 5 | 5 | 5 | 5 | 5 | 7 | 3 | 6 |

*Classic graph data for graph classification.* We summarize the involved TUDdataset in the following Table 3.

Table 3: TUDataset description.

|  | **MUTAGE** | **NCI1** | **ENZYMES** | **D&D** | **PTC_FM** | **IMDB** | **PROTEINS** |
|---|---|---|---|---|---|---|---|
| **#Graphs** $|\mathcal{V}|$ | 188 | 4,110 | 600 | 1,178 | 349 | 1,000 | 1,113 |
| **#Classes** $|\mathcal{P}|$ | 2 | 2 | 6 | 2 | 2 | 2 | 2 |
| **Avg#Nodes** $|\mathcal{Y}|$ | 17.93 | 29.87 | 32.63 | 284.32 | 14.11 | 19.77 | 39.06 |
| **Avg#Edges** $|\mathcal{Y}|$ | 19.79 | 32.30 | 62.14 | 715.66 | 14.48 | 96.53 | 72.82 |

*Disease-based human connectome data.* We summarize the data information in the following Table 4. Note, Destrieux atlas [10] (160 brain regions) are used in OASIS to verify the scalability of the models.

Table 4: Disease-based human connectome data statistics.

| Dataset | Condition | # of Subjects | # of Classes | # of Regions/Nodes | Avg # of Node Features |
|---|---|---|---|---|---|
| **ABIDE** | Autism | 1025 | 2 | 116 | 201 |
| **ADNI** | Alzheimer | 250 | 5 | 116 | 177 |
| **OASIS** | Alzheimer | 1475 | 2 | 160 | 330 |
| **PPMI** | Parkinson | 209 | 4 | 116 | 198 |

For a binary dataset consisting of two classes, one representing a disease group and the other a normal control group. For ADNI dataset, following the clinical outcomes, we categorized subjects into distinct groups representing different cognitive statuses. These groups include: cognitive normal (CN), Subjective memory concern (SMC), early-stage mild cognitive impairment (EMCI), late-stage mild cognitive impairment (LMCI), and Alzheimer's Disease (AD) groups. To facilitate population counts, we regard CN, SMC and EMCI as "CN-like" group, while LMCI and AD as "AD-like" groups. This partitioning allows for the analysis and comparison of individuals across varying levels of cognitive function, providing valuable insights into disease progression and cognitive decline within the study population. For the PPMI dataset, which encompasses four distinct classes, including normal control, scans without evidence of dopaminergic deficit (SWEDD), prodromal, and Parkinson's disease (PD).

*Hyperparameters.* We use the Adam optimizer with a learning rate of 0.01, and the epoch is set as 250. Most hidden dimensions are set to 128 (Squirrel and Chameleon are set to 64, Cora and ABIDE are set to 32). The code is released at GitHub: `https://github.com/Dandy5721/Game-of-GNN`.

## B.2 Comparison Methods

Graph neural networks (GNNs) have emerged as powerful tools for learning from graph-structured data, achieving state-of-the-art performance in various domains such as social networks, biological networks, and recommendation systems. In this work, we compare our method against a diverse set of benchmark GNN models that represent key advancements in the field:

Vanilla GCN [17]: The Graph Convolutional Network (GCN) introduced the foundational concept of convolutional operations on graph-structured data, leveraging spectral graph theory to propagate node features across the graph. Despite its simplicity, GCN remains a widely used baseline in GNN research.

GAT [34]: The Graph Attention Network (GAT) improved upon GCN by incorporating an attention mechanism to adaptively weigh neighbor contributions, enabling the model to capture more nuanced patterns in heterogeneous and large-scale graphs.

GraphSAGE [13]: This inductive framework generates embeddings by sampling and aggregating features from node neighborhoods, making it particularly effective for large and dynamic graphs where new nodes can be introduced.

GraphCON [30]: GraphCON leverages neural ODEs and skip connections to improve gradient flow during training, enabling it to address the oversmoothing problem in deep GNN architectures.

GraphBel [31]: This model focuses on enhancing robustness against adversarial attacks by learning more resilient graph representations through belief propagation mechanisms.

GRAND [7]: GRAND introduces random diffusion processes to improve message passing, focusing on long-range dependencies and reducing the oversquashing issue commonly seen in deep GNNs.

ACMP [36]: The Adversarial Contrastive Message Passing (ACMP) framework utilizes contrastive learning to enhance node representations, particularly in the presence of noisy or incomplete graphs.

HANG [42]: HANG employs adversarial training to learn robust graph embeddings, effectively tackling challenges posed by graph perturbations and adversarial noise.

GIN [40]: The Graph Isomorphism Network (GIN) is designed to be as powerful as the Weisfeiler-Lehman graph isomorphism test, achieving high expressiveness by using a learnable aggregation function.

AM-GCN [35]: This model integrates both node features and graph topology in a balanced way, enhancing its ability to learn from graphs with highly diverse connectivity patterns.

These models collectively capture a wide range of design principles, from improved aggregation mechanisms (e.g., GAT, GIN) and inductive capabilities (e.g., GraphSAGE) to adversarial robustness (e.g., HANG, GraphBel) and advanced training techniques (e.g., GraphCON, GRAND). By benchmarking against these state-of-the-art GNNs, we provide a comprehensive evaluation of our method's performance, highlighting its strengths and contributions to the field.

## C   Discussion and Limitations

*Discussion.* To systematically investigate the impact of the ICNN depth on model performance, we conducted a comprehensive ablation study with ICNN depths of $\{2, 4, 8, 16, 32, 64\}$ across nine benchmark datasets. Table 5 reports both the mean test accuracy and the corresponding $\alpha$ for each configuration, with homophily ratios ($h$) indicated in parentheses.

Two major trends emerge from these ablation results. First, datasets with smaller homophily ratios ($h$) generally require deeper ICNN architectures to achieve satisfactory performance. This observation suggests that heterophilic graphs exhibit more complex and non-local dependencies between connected nodes, thereby necessitating deeper neural representations to effectively capture the underlying feature–structure relationships. Second, the scaling factor $\alpha$ is primarily determined by the homophily ratio $h$ and remains largely insensitive to network depth. In other words, the intrinsic coupling between $\alpha$ and $h$ persists regardless of ICNN depth, indicating that $\alpha$ reflects structural characteristics of the underlying graph rather than architectural complexity. Taken together, our findings demonstrate that while deeper ICNN modules can better capture the intricate edge

Table 5: Ablation results on ICNN depth. Each entry reports mean test accuracy (%) $\pm$ standard deviation, along with the corresponding $\alpha$.

| Depth | Texas (0.11) | Wisconsin (0.21) | Squirrel (0.22) | Actor (0.22) | Chameleon (0.23) | Cornell (0.30) | Citeseer (0.57) | Pubmed (0.74) | Cora (0.81) |
|---|---|---|---|---|---|---|---|---|---|
| 2 | $86.23 \pm 3.2$ | $88.22 \pm 4.3$ | $37.98 \pm 0.7$ | $32.82 \pm 0.27$ | $57.77 \pm 1.5$ | $85.76 \pm 5.6$ | $75.17 \pm 8.6$ | $86.33 \pm 1.0$ | $87.53 \pm 0.7$ |
| 4 | $84.40 \pm 11.8$ | $88.44 \pm 5.5$ | $39.32 \pm 0.7$ | $32.53 \pm 0.40$ | $61.26 \pm 1.8$ | $83.59 \pm 2.2$ | $77.61 \pm 1.4$ | $90.75 \pm 0.3$ | $87.80 \pm 0.5$ |
| 8 | $83.79 \pm 11.3$ | $84.44 \pm 8.3$ | $38.18 \pm 1.2$ | $36.28 \pm 1.4$ | $62.48 \pm 15.5$ | $83.93 \pm 4.5$ | $77.96 \pm 4.1$ | $82.96 \pm 1.3$ | $87.09 \pm 0.8$ |
| 16 | $85.32 \pm 3.7$ | $87.17 \pm 1.8$ | $40.38 \pm 0.6$ | $36.69 \pm 6.2$ | $60.14 \pm 0.2$ | $83.32 \pm 3.7$ | $73.82 \pm 6.9$ | $84.59 \pm 1.4$ | $87.21 \pm 0.7$ |
| 32 | $90.12 \pm 5.6$ | $88.44 \pm 4.2$ | $41.12 \pm 0.9$ | $34.84 \pm 5.4$ | $63.85 \pm 0.9$ | $83.01 \pm 6.7$ | $73.12 \pm 6.4$ | $87.40 \pm 0.5$ | $86.04 \pm 0.9$ |
| 64 | $94.00 \pm 2.2$ | $90.66 \pm 4.1$ | $37.39 \pm 0.5$ | $33.55 \pm 4.6$ | $59.43 \pm 15.2$ | $81.65 \pm 11.1$ | $65.46 \pm 5.6$ | $83.03 \pm 1.0$ | $85.10 \pm 0.6$ |
| | | | | | $\alpha$ | | | | |
| 2 | 1.1345 | 0.8909 | 0.8661 | 0.8425 | 0.8129 | 0.7041 | 0.4578 | 0.1315 | 0.1153 |
| 4 | 1.1476 | 1.0462 | 0.9514 | 0.8815 | 0.7616 | 0.6660 | 0.3623 | 0.1312 | 0.1076 |
| 8 | 1.0669 | 1.0188 | 0.9431 | 0.9032 | 0.7391 | 0.6918 | 0.5043 | 0.1426 | 0.1103 |
| 16 | 1.1607 | 1.0356 | 0.9496 | 0.9006 | 0.7856 | 0.6487 | 0.4198 | 0.1310 | 0.1091 |
| 32 | 1.0551 | 0.9312 | 0.8848 | 0.8781 | 0.8283 | 0.6767 | 0.3757 | 0.1422 | 0.1111 |
| 64 | 1.2591 | 1.1405 | 1.1089 | 0.9013 | 0.7025 | 0.6563 | 0.4546 | 0.1428 | 0.1157 |

relationships present in heterophilic data, the learned $\alpha$ parameter remains a stable descriptor of graph homophily, independent of the model's depth.

*Limitations.* Although *Game-of-GNN* exhibits slightly higher inference time on small graphs (e.g., 0.0331 s on Cora compared with 0.0027 s for GCN), its performance on large-scale datasets reveals an unexpected yet encouraging trend Table 6. On the OGBN-PRODUCTS dataset ($\sim$2.45 M nodes), *Game-of-GNN* achieves an inference time of 0.2971 s, which is comparable to that of GCN (0.2700 s), despite its dynamical system formulation. In contrast, other dynamical GNN architectures such as GraphBel, ACMP, GraphCON, and HANG encounter out-of-memory (OOM) errors under the same data loading protocol (PygNodePropPredDataset). All experiments were conducted on a system equipped with six NVIDIA RTX 6000 Ada GPUs (48 GB each), and no subgraph-based mini-batching techniques were employed.

This observation can be attributed to three key factors. (1) *Lightweight iterative solver.* Although *Game-of-GNN* performs step-wise integration of a learned PDE system via an explicit Euler scheme, each update step is computationally inexpensive and involves only local, sparse neighborhood operations—analogous to the message passing in GCN. Unlike attention-based models such as GAT or GraphBel, our update rule consists of deterministic flow steps with a fixed per-node computational cost. (2) *Effective hardware utilization on large graphs.* On small graphs, GPU resources are underutilized, making even minor per-node overheads (e.g., from PDE state updates) more noticeable. On large graphs, however, the abundant computational workload fully occupies the GPU, thereby amortizing the integration cost across many nodes and yielding inference times close to GCN.

Table 6: Training and inference time comparison (in seconds per epoch and milliseconds per inference) across datasets. "OMM" indicates out of memory.

| Model | Cora (2,708 nodes) | | Pubmed (19,717 nodes) | | OGBN-ARXIV ($\sim$170K) | | OGBN-PRODUCTS ($\sim$2.45M) | |
|---|---|---|---|---|---|---|---|---|
| | Train (s/ep) | Infer (ms) | Train (s/ep) | Infer (ms) | Train (s/ep) | Infer (ms) | Train (s/ep) | Infer (ms) |
| GCN | 0.0098 | 0.0027 | 0.0097 | 0.0046 | 0.0147 | 0.0076 | 0.3602 | 0.2700 |
| GAT | 0.0201 | 0.0050 | 0.0202 | 0.0053 | 0.0779 | 0.0319 | OMM | OMM |
| GraphSAGE | 0.0059 | 0.0015 | 0.0064 | 0.0016 | 0.0184 | 0.0071 | 0.4149 | 0.1757 |
| GraphCON | 0.0394 | 0.0026 | 0.2044 | 0.0177 | 0.0900 | 0.0480 | OMM | OMM |
| GraphBel | 0.1943 | 0.0241 | 0.2022 | 0.0326 | 0.7426 | 0.1638 | OMM | OMM |
| ACMP | 0.2172 | 0.0288 | 0.3678 | 0.0635 | 0.7193 | 0.1970 | OMM | OMM |
| HANG | 0.0794 | 0.0335 | 0.1103 | 0.0417 | 0.1638 | 0.0951 | OMM | OMM |
| *Game-of-GNN* | 0.0755 | 0.0331 | 0.0924 | 0.0324 | 0.1023 | 0.0376 | 0.8145 | 0.2971 |

# D  Impact Statement

Our major contribution to the machine learning field is that we introduce a principled approach to optimize GNNs for fitting diverse graph datasets. Through the integration of mean-field control theory and Hamiltonian flows into GNN abstract learning, we developed a novel methodology that enhances our understanding of deep learning models applied to graph datasets.

From the application perspective, our deep model represents a promising approach to bridge the gap between graph-based machine learning and neuroscience research, offering new avenues for studying disease processes.

