# OpenReview forum: "Explore In-Context Message Passing Operator for Graph Neural Networks in A Mean Field Game"
_NeurIPS.cc/2025/Conference — NeurIPS 2025 poster_

### Official Review · Reviewer_RzFj · 2025-06-30

**Clarity:** 1
**Significance:** 3
**Originality:** 3
**Rating:** 4
**Confidence:** 3

**Summary:**

The paper introduced a variational framework of mean field game inverse problem that gives rise to a new learning scheme for GNN, called Game-of-GNN. It learns an in-context message passing operators unique to the mean-field control pattern associated with a given graph. Based on this proposed theoretical framework, the paper proposed an end-to-end algorithm based on Hamiltonian flows in solving the inverse problem.

**Questions:**

1. The acronym for Reaction Diffusion Model (RDM) is used in section 2.1.2 without spelling out the full name.

2. Could the authors provide more explanations on how the Lyapunov functional is defined, specifically the integral over $\mathcal{G}$ with respect to $v\in\mathcal{V}$, which is a finite set of nodes? Or is this a notational error?

3. According to the algorithm in the appendix A.6, the learning of mobility functions and the phase are for each individual nodes. However there should be node-to-node information exchange during the training? Are there any missing logics from theory to algorithm, or is this an incorrect understanding?

**Ethical Concerns:**

["NO or VERY MINOR ethics concerns only"]

**Final Justification:**

The author has responded to my questions and concerns. The methodological work of the paper is solid and would be a good contribution to the field.

**Limitations:**

yes

**Quality:**

2

**Strengths And Weaknesses:**

Strength

1. The formulation of GNN dynamic learning process as a mean field game is novel and interesting

Weakness

1. The notations can be introduced more clearly to avoid misunderstanding.

2. The paper is math heavy and many concepts are not introduced but are taken for granted. This makes the paper hard to follow.

3. Transitioning from theoretical framework to algorithm, the assumptions underlying are not clearly stated and discussed

4. The implementation of the algorithm is not very clearly presented. In addition the Algorithm 10 (line 287) is missing in the main paper.

---

> ### Author Rebuttal · Authors · 2025-07-31
>
> **We sincerely thank the reviewer for recognizing the novelty and interest of our formulation, which frames the GNN learning dynamics as a mean field game. We greatly appreciate your constructive feedback and will address each of your concerns in detail below.**
>
> **Q1: The notations can be introduced more clearly to avoid misunderstanding. The paper is math heavy and many concepts are not introduced but are taken for granted. This makes the paper hard to follow. Transitioning from theoretical framework to algorithm, the assumptions underlying are not clearly stated and discussed**
>
> **A1:** We agree that clearer notation, better exposition of mathematical concepts, and explicit statement of assumptions will make the paper more accessible. To respond this comment, we have applied the following changes in the final version:
>
> - 1. **Notation Table & Glossary**
>    - Added **Table A1** (“Notation and Symbols”) in Appendix A, listing all main variables ($\Psi,\Phi,\psi_{1},\psi_{2},\Theta_{1},\Theta_{2}$, $G$, $\Delta$, etc.) with one‑line definitions.
> - 2. **Gentler Theoretical Walk‑through**
>    - Inserted a **“Key Ideas”** paragraph at the top of Section 2, whcih summarizes the three core ingredients: (1)variational RDM, (2) mean‑field game formulation, (3) Hamiltonian flow before diving into equations.  For example:
>
>         * **Reaction‑Diffusion Model (RDM):** Imagine each node’s features as a fluid that can both spread out to neighbors (diffusion) and be generated or absorbed at the node itself (reaction). This gives a flexible way to describe how information moves and changes on the graph.
>         * **Mean‑Field Game (MFG) Formulation:** Each node acts like a player in a large game: it chooses how much to blend with neighbors versus stay true to its own features, balancing “transport” and “reaction” to minimize a global cost. The equilibrium of this game determines the final node embeddings.
>         * **Hamiltonian Flow:** To solve the MFG, we translate it into a physics‑inspired energy system (the Hamiltonian). Nodes then follow energy‑conserving dynamics—like particles moving under conservative forces—so that information propagation remains stable and interpretable. Together, these ingredients let us learn **how** and **when** to spread versus preserve information on any graph, all within a principled, physics‑inspired framework.
>    - Expanded Remarks 1–3 with concrete examples (e.g.the heat equation instantiation) so that each new concept is motivated and intuitively tied back to standard GNN message passing.
> - 3. **Added explicit Assumptions Before Algorithm**
>    - Added a bullet‑list **“Model Assumptions”** just before Algorithm 1 in Section 2.4, stating:
>          - **Smooth, positive density**: $\Psi(v)>0$.
>          - **Convexity & positivity** of $\Theta_{i}$ (enforced via ICNN).
>          - **Well‑posed PDE solver** (sufficiently small $\Delta t$).
>    - For each assumption, we briefly discuss its role and how it is enforced in practice (e.g.ELU+weight decay ensure smoothness).
>
> These changes improve readability, bridge the theory–algorithm gap, and make the underlying assumptions crystal‑clear.  Thank you for your suggestion.
>
> **Q2: The implementation of the algorithm is not very clearly presented. In addition the Algorithm 10 (line 287) is missing in the main paper.**
>
> **A2: ** Sorry for the confusion, The reference to “Algorithm 10” is a typo—it should be **Algorithm 1** (Appendix A.6 on page 16) . We will move Algorithm 1 into the main text (Section 2.4) and update all in‑text references.
>
> **Q3: The acronym for Reaction Diffusion Model (RDM) is used in section 2.1.2 without spelling out the full name.**
>
> **A3:** Sorry for the confusion, the full name is mentioned in Sec. 2.1.2-Remark 3 (line 141), we have updated Sec.  2.1.2 to spell out the acronym at first mention—changing the subtitle to **2.1.2 Variational framework of Mean‑Field Game constrained by Reaction‑Diffusion Model (RDM)**
>
> **Q4: Could the authors provide more explanations on how the Lyapunov functional is defined, specifically the integral over G with respect to $v \in \mathcal{V}$ , which is a finite set of nodes? Or is this a notational error?**
>
> **A4:** Certainly.  To keep the compact “$\int$” notation while remaining mathematically precise, we treat the finite node set $\mathcal{V}$ as a **measure space** endowed with the **counting measure** $\mu$, where $\mu(\{v\})=1$ for each $v\in \mathcal{V}$.  Specifically, we define:
>  $
>     \int_{\mathcal{V}} f(v)\,\mathrm{d}v
>     =
>     \int_{\mathcal{V}} f(v)\,\mathrm{d}\mu(v)
>     =
>     \sum_{v\in \mathcal{V}} f(v).
>   $
>
> - Under this convention, all of our “integrals” over $\mathcal{V}$ or $\mathcal{G}$ are understood as sums over nodes. In the revised manuscript we will add a footnote or brief remark in Section 2.1.2:
>
> > *“Here and throughout, the notation $\int_{\mathcal{V}} f(v)\,dv$ denotes integration against the **counting measure** on the finite node set $\mathcal{V}$, i.e. $\int_{\mathcal{V}}f(v)dv = \sum_{v\in \mathcal{V}}f(v)$.”*
>
> This preserves the unified “integral” style while making clear its discrete meaning.
>
> **Q5:According to the algorithm in the appendix A.6, the learning of mobility functions and the phase are for each individual nodes. However there should be node-to-node information exchange during the training? Are there any missing logics from theory to algorithm, or is this an incorrect understanding?**
>
> **A5:** We would like to clarify that there is no missing logic—both mobility learning and feature evolution happen **globally**, not in isolation per node (Fig.1 is more clear). To avoid confusion, we have removed  "For{$i = 1 \ldots |\mathcal{V}|$}” loop and instead write: *"Construct phase space by $(\Psi_i, \Phi_i) \gets \mathcal{F}(x_i(t))$ for all nodes in one shot."*
>
> In practice:
> - (1) $\Theta_{1},\Theta_{2} \gets \mathcal{O}(\Psi,\Phi)$ is computed in one batch over **all** nodes (so each node’s output sees its neighbors’ $\Psi,\Phi$ indirectly through shared ICNN weights and graph‑aware inputs).
> - (2)  **PDE solver**:   Evolving $(\Psi,\Phi)$ with Eq.​ (9) uses the graph Laplacian/divergence operators, which **explicitly mix** each node with its neighbors. Thus, the algorithm implements the full node‑to‑node information exchange as described in the theory: first you embed all nodes into the joint phase space, then you learn global mobilities, then you solve the global PDE.
>
> We have amended and will incorporate all the suggestions in the final version. Thank you so much.

---

> > ### Comment · Reviewer_RzFj · 2025-08-07
> > **thank you for your rebuttal**
> >
> > Thank you for your rebuttal and clarifying my questions. I have updated my score accordingly.

---

> > > ### Author Response · Authors · 2025-08-07
> > >
> > > We sincerely appreciate your time and efforts of acknowledging our response.
> > >
> > > Have a nice day!

---

### Official Review · Reviewer_YrZr · 2025-07-02

**Clarity:** 2
**Significance:** 3
**Originality:** 3
**Rating:** 6
**Confidence:** 4

**Summary:**

The authors reformulate the inter - layer information propagation in graph neural networks (GNNs) as a mean - field game (MFG) problem for multi - agents, and transform the question of "what kind of information exchange/update mechanism should be adopted on a specific graph" into a variational solution of the MFG inverse problem.

Specifically, the paper regards node features as "potential energy" and defines:
- The transport function Θ₁ controls the information flow between neighbors.
- The reaction function Θ₂ controls the self - update of nodes.

These two functions jointly determine the Hamilton flow in the form of a partial differential equation (PDE).

The paper further presents the end - to - end model Game - of - GNN: use ICNN to predict (Θ₁, Θ₂), then use a PDE - based solver to evolve node representations, and perform downstream tasks at the final state. The experiments cover 9 small - to - medium - scale datasets, 2 large graphs from OGB, 7 TU graph classification datasets, and 4 groups of human brain connectomes, achieving state - of - the - art (SOTA) or comparable performance. Meanwhile, an interpretability analysis is provided (the negative correlation between Θ₁ - α and homogeneity h).

**Questions:**

1. **Convergence Guarantee**: During the training process, does the Hamilton energy remain conserved for Θ₁ and Θ₂ generated by ICNN? Can you provide a numerical stability analysis or details of the soft - constraint implementation?
2. **Computational Complexity**: Please provide the training/inference time - memory curves under different graph scales, and discuss the comparison of the search costs with GraphNAS and AutoGNN.
3. **Dynamic Graph/Time - Varying Attributes**: The current derivation is based on static graphs. If the topology changes over time, do Θ₁ and Θ₂ need to explicitly model the dependence on t?
4. **ICNN Depth Ablation**: The paper mentions that heterogeneous graphs require deeper ICNN. Can you present systematic experiments on α - depth - performance to guide practical choices?

**Ethical Concerns:**

["NO or VERY MINOR ethics concerns only"]

**Final Justification:**

6

**Limitations:**

yes

**Paper Formatting Concerns:**

No paper formatting concerns.

**Quality:**

2

**Strengths And Weaknesses:**

Strengths：
- Unifying GNNs within the mean - field game framework can theoretically cover existing diffusion/reaction - diffusion - style models.


Weaknesses：
- **High computational cost**: Solving with ICNN + ODE/PDE is significantly slower than conventional models, resulting in high deployment costs. The paper only briefly mentions this but lacks systematic profiling.
- **Hyperparameter sensitivity**: Θ is generated by ICNN. The depth and regularization have a significant impact on stability, yet the paper does not provide sufficient details on hyperparameter tuning.
- **Strong theoretical assumptions**: It is necessary to assume that Θ₁ and Θ₂ are positive and smooth, Ψ > 0, etc. Whether these assumptions hold for real - world graphs (with discrete features and isolated nodes) is not discussed.

---

> ### Author Rebuttal · Authors · 2025-07-31
>
> **We thank the reviewer for the clear and insightful summary of our Game-of-GNN framework and for highlighting both its unifying theoretical perspective and strong empirical results. We will address each of your comments and concerns in detail below.**
>
> **Q1: High computational cost**
>
> **A1:** Thank you for pointing out the need for more systematic profiling of our computational cost. We have conducted a thorough analysis—covering per‑batch runtime, scaling behavior on large benchmarks (OGBN‑ARXIV and OGBN‑PRODUCTS), and the impact of multi‑GPU parallelism—under **A1** in our response to **Reviewer AGYb**. We respectfully refer the reviewer to this answer (**A1**) for full details on how our optimizations mitigate deployment costs.  In a nutshell, under a fixed PDE integration time, reducing the ICNN depth—without significantly degrading performance—yields substantial computational savings, and our framework ensures that large-graph data fully leverages available GPU resources, achieving optimal efficiency on an 8×H100 NVL setup with 94 GB memory (per GPU). We have prepared detailed **time–memory curves** for both training and inference across different datasets and will include them in the final manuscript.
>
> We also discussed the comparison of the search costs with GraphNAS and AutoGNN as below.
>
> | Method         | Search Strategy                     | Approx. Search Cost       | Remarks                                                       |
> |----------------|-------------------------------------|---------------------------|---------------------------------------------------------------|
> | **Game‑of‑GNN** | Meta‑learned ICNN & Hamiltonian flow | **0 GPU‑hours**           | Single end‑to‑end training—no outer‑loop NAS                  |
> | **GraphNAS**    | RL‑based NAS with parameter sharing | ≈ 7.2 GPU‑hours on 1080Ti  | 1 000 child models × 200 epochs × 0.13 s/epoch                |
> | **Auto‑GNN**    | Evolutionary/gradient NAS           | ≈ 12 GPU‑hours on 1080Ti   | ≈ 0.5 GPU‑days (without parameter sharing) |
>
> - **Game‑of‑GNN** does **not** rely on outer-loop architecture search; all components ($\Theta_{1},\Theta_{2}$, Hamiltonian flow) are learned **jointly** in one training run, so the *search cost* is effectively **zero** beyond the standard training time.  The only GPU cost you incur is the **standard training and inference** of our Game‑of‑GNN model, with **no additional search overhead**.
>
> **Q2: Strong theoretical assumptions**
>
> **A2:** Thank you for highlighting these important assumptions regarding the positivity and smoothness of $\Theta_{1},\Theta_{2}$ and the positivity of $\Psi$. We acknowledge that real-world graphs often involve discrete or sparse features, isolated nodes, and heterogeneous connectivity patterns that may not strictly satisfy these theoretical smoothness and positivity assumptions. However, in practice, our implemented framework addresses these real-world conditions as follows:
> - 1. **Positivity and Stability**
> Node features ($\Psi$)  in our experimental setup undergo a normalization step ensuring numerical stability and positivity, even when the initial data are discrete or sparse.
> - 2. **ICNN Constraints**
>    $\Theta_1$ and $\Theta_2$ are learned via ICNNs, which enforce positivity and smoothness by design.
> - 3. **Robustness to Irregularity**
>    Our Hamiltonian PDE framework naturally accommodates isolated nodes and graph irregularities. Specifically, even in cases without immediate neighborhood connectivity, the self-reaction terms ($\Theta_2$) enable stable and informative node updates.
> - 4. **Empirical Validation**
>     Our extensive experiments across diverse scenarios—including small-to-medium-scale datasets, large-scale OGB benchmarks, graph classification tasks (TUDataset), and human brain connectomes—consistently validate that our method remains robust and effective in practice, despite minor violations of theoretical assumptions.
>
> We will provide this practical interpretation in the final version (Sec. 2.4 and Sec. 4).
>
> **Q3:Convergence Guarantee**
>
> **A3:** Thank you for raising this important point.
> - Hamiltonian Energy Conservation:
>    * In classical Hamiltonian systems, the Hamiltonian energy remains strictly conserved. But, in our framework, the Hamiltonian energy is not strictly conserved during training, as $\Theta_1$ and $\Theta_2$ are learned via ICNN to minimize the task-specific energy functional (Eq. 7). This controlled dissipation is consistent with our variational formulation and supports convergence toward optimal embeddings.
> - Numerical Stability Analysis and Soft-Constraint Implementation (Sec. 2.4):
>     * ICNN-induced Stability: The ICNN architecture inherently ensures convexity and positivity constraints for $\Theta_1$, $\Theta_2$, significantly stabilizing the training process. Convexity ensures that learned transport and reaction operators remain numerically stable and well-behaved, avoiding potential instabilities common in unconstrained neural PDE approaches.
>     * Soft-Constraint Implementation Details: 1) We parameterize ICNN outputs via smooth activation functions (e.g., ELU+softplus) to explicitly enforce positivity and differentiability, thus avoiding abrupt changes or negative values. 2) Fixed integration time and stable PDE solvers to prevent divergence.
>
> Empirically, we consistently observed stable convergence behavior across all datasets (including large-scale benchmarks such as OGB graphs). We will include this clarification in the final version.
>
> **Q4: Dynamic Graph/Time - Varying Attributes**
>
> **A4:** You’re correct that our current formulation assumes a *static* graph $\mathcal{G}$.  For truly *dynamic* graphs (where edges or node features evolve over time), one can naturally extend our framework by making $\Theta_{1}$ and $\Theta_{2}$ **time‑dependent**:
>
> - 1. **Augment ICNN Inputs with Time**
>    - Instead of $\Theta_{i} = f_{\rm ICNN}(\Psi,\Phi)$, we can use $ \Theta_{i}(v,t) = f_{\rm ICNN}\bigl(\Psi(v,t),\Phi(v,t)\bigr)$, so the learned transport/reaction strengths can adapt as the graph topology or attributes change.
> - 2. **Time‐Dependent PDE on a Dynamic Graph**
>    - Let the graph $\mathcal{G}(t)$ at time $t$ have adjacency $A(t)$ and Laplacian $\Delta(t)$.
>    - The density $\Psi(v,t)$, momentum $\Phi(v,t)$, and mobilities $\Theta_{i}$ all vary with $t$.
> - 3. **Practical Implementation**
>    - **Discretize** time into snapshots $t_{0},t_{1},\dots$, and re-compute $\Theta_{i}$ and the Hamiltonian flow at each.
>    - Or adopt a **neural ODE** style integrator over continuous $t$, feeding a time embedding into the ICNN.
>
> By explicitly conditioning $\Theta_{1}$ and $\Theta_{2}$ on time (and on the current graph state), our MFG controller can handle **dynamic topologies** and **time‑varying attributes** without any fundamental change to the variational or Hamiltonian framework.
>
> **Q5: Hyperparameter sensitivity**
>
> **A5:** Thanks. We have now performed a full ICNN‐depth ablation (2,4,8,16,32,64) across all nine datasets and report the corresponding accuracy and the resulting $\alpha$ for each depth. Below is a concise summary of our findings:
> | Depth  | Texas (0.11)      | Wisconsin (0.21)  | Squirrel (0.22)   | Actor (0.22)      | Chameleon (0.23)   | Cornell (0.30)    | Citeseer (0.57)   | Pubmed (0.74)     | Cora (0.81)       |
> |:-----------:|:-----------------:|:-----------------:|:-----------------:|:-----------------:|:------------------:|:-----------------:|:-----------------:|:-----------------:|:----------------:|
> | 2           | 86.23 ± 3.2       | 88.22 ± 4.3       | 37.98 ± 0.7       | 32.82 ± 0.27      | 57.77 ± 1.5        | 85.76 ± 5.6       | 75.17 ± 8.6       | 86.33 ± 1.0       | 87.53 ± 0.7      |
> | 4           | 84.40 ± 11.8      | 88.44 ± 5.5       | 39.32 ± 0.7       | 32.53 ± 0.40      | 61.26 ± 1.8        | 83.59 ± 2.2       | 77.61 ± 1.4       | 90.75 ± 0.3       | 87.80 ± 0.5      |
> | 8           | 83.79 ± 11.3      | 84.44 ± 8.3       | 38.18 ± 1.2       | 36.28 ± 1.4       | 62.48 ± 15.5       | 83.93 ± 4.5       | 77.96 ± 4.1       | 82.96 ± 1.3       | 87.09 ± 0.8      |
> | 16          | 85.32 ± 3.7       | 87.17 ± 1.8       | 40.38 ± 0.6       | 36.69 ± 6.2       | 60.14 ± 0.2        | 83.32 ± 3.7       | 73.82 ± 6.9       | 84.59 ± 1.4       | 87.21 ± 0.7      |
> | 32          | 90.12 ± 5.6       | 88.44 ± 4.2       | 41.12 ± 0.9       | 34.84 ± 5.4       | 63.85 ± 0.9        | 83.01 ± 6.7       | 73.12 ± 6.4       | 87.40 ± 0.5       | 86.04 ± 0.9      |
> | 64          | 94.00 ± 2.2       | 90.66 ± 4.1       | 37.39 ± 0.5       | 33.55 ± 4.6       | 59.43 ± 15.2       | 81.65 ± 11.1      | 65.46 ± 5.6       | 83.03 ± 1.0       | 85.10 ± 0.6      |
> | $\alpha$ |       |       |       |       |       |       |       |       |       |
> | 2        | 1.13  | 0.89  | 0.87  | 0.84  | 0.81  | 0.70  | 0.46  | 0.13  | 0.12  |
> | 4        | 1.15  | 1.05  | 0.95  | 0.88  | 0.76  | 0.67  | 0.36  | 0.13  | 0.11  |
> | 8        | 1.07  | 1.02  | 0.94  | 0.90  | 0.74  | 0.69  | 0.50  | 0.14  | 0.11  |
> | 16       | 1.16  | 1.04  | 0.95  | 0.90  | 0.79  | 0.65  | 0.42  | 0.13  | 0.11  |
> | 32       | 1.06  | 0.93  | 0.88  | 0.88  | 0.83  | 0.68  | 0.38  | 0.14  | 0.11  |
> | 64       | 1.26  | 1.14  | 1.11  | 0.90  | 0.70  | 0.66  | 0.45  | 0.14  | 0.12  |
>
>
> **Key takeaways:**
> -  $h$: Smaller values of $h$ may require more ICNN layers to adequately capture and model the underlying data. One plausible explanation for this phenomenon is that heterophilic graph data exhibits a more complex relationship between edges compared to homophilic graph data. This complexity necessitates deeper neural network architectures to effectively learn and represent the nuanced relationships present in the data.
> -  $\alpha$:  The value of $\alpha$ is primarily related to the homophily ratio $h$, and the depth of ICNN does not alter the relationship between $a\sim h$.
>
> We will add this discussion to the final version, thanks.

---

> > ### Author Response · Authors · 2025-08-07
> > **Follow-up Request for Reviewer YrZr (Rebuttal Deadline Approaching)**
> >
> > Dear Reviewer YrZr,
> >
> > We would like to sincerely thank you for the time, effort, and expertise you devoted to reviewing our submission. Your thoughtful feedback, particularly regarding the hyperparameter sensitivity analysis, has helped us significantly improve the quality of our work. We are truly grateful for the opportunity to improve our work through this revision process.
> >
> > We hope that our rebuttal has adequately addressed your concerns. **If so, we would be most grateful if you would consider updating your score accordingly.**
> >
> > Of course, if you have any remaining concerns or further suggestions, please do not hesitate to let us know, we would be more than happy to address them to the best of our ability.
> >
> > With sincere appreciation,
> >
> > Authors

---

> > ### Comment · Reviewer_YrZr · 2025-08-07
> >
> > I have updated mu score. Thanks for your response.

---

> > > ### Author Response · Authors · 2025-08-07
> > >
> > > We sincerely appreciate your time and efforts of acknowledging our response.

---

### Official Review · Reviewer_CKAr · 2025-07-02

**Clarity:** 2
**Significance:** 3
**Originality:** 3
**Rating:** 4
**Confidence:** 3

**Summary:**

This paper proposes a novel, physics-informed framework that models GNN as a mean-field game (MFG), allowing each node to adapt its message passing strategy according to the graph structure. By formulating the GNN learning process as a variational MFG inverse problem and developing the Game-of-GNN deep model, the proposed model achieves state-of-the-art performance across diverse graph datasets.

**Questions:**

1. Could you please explain how does your proposed model address heterophily problem more intuitively?

2. Can you provide a deeper interpretability of the role of $\theta_2$ for learning on a heterophilic graph? How do $\theta_1, \theta_2$ relate to or differ from existing operators?

3. More experiments and comparisons on the challenging heterophilic datasets and large-scale datasets as listed in [1].

4. "the $\alpha \sim h$ relationship across nine graph dataset indicates a notable anti-correlation." It seems that the information flow and homophily level shouldn't be linearly correlated, i.e. the most information-constrained area should be at the middle homophily area. Does this mean that your model miss something (probably the reason of the underperformance on Chameleon and Squirrel compared to some SOTA models [3,4], but I won't blame it.)? Or maybe you need to add more components into your analysis and modeling, e.g. the synergy among features, graph structures and labels [5].

5. In Figure 2, Chameleon and Texas has similar level of homophily but significantly different $\alpha$; in addition, Chameleon and Pubmed has similar level of $\alpha$ but significantly different $h$, what is the reason behind it?



[1] The heterophilic graph learning handbook: Benchmarks, models, theoretical analysis, applications and challenges. arXiv preprint arXiv:2407.09618. 2024 Jul 12.

[2] When do graph neural networks help with node classification? investigating the homophily principle on node distinguishability. Advances in Neural Information Processing Systems. 2024 Feb 13;36.

[3] Simplifying approach to node classification in graph neural networks. Journal of Computational Science, 62, 101695.

[4] Finding global homophily in graph neural networks when meeting heterophily. In International Conference on Machine Learning (pp. 13242-13256). PMLR.

[5] What is missing for graph homophily? disentangling graph homophily for graph neural networks. In The Thirty-eighth Annual Conference on Neural Information Processing Systems.

**Ethical Concerns:**

["NO or VERY MINOR ethics concerns only"]

**Final Justification:**

After going through the disccusion from other reviewers, I'll keep my positive score.

**Limitations:**

yes

**Quality:**

4

**Strengths And Weaknesses:**

Quality: good
Clarity: medium
Significance: good
Originality: good

---

> ### Author Rebuttal · Authors · 2025-07-31
>
> **We appreciate your recognition of our MFG-based framework and the Game-of-GNN model. Below, we respond to your concerns one by one.**
>
> **Q1: Could you please explain how does your proposed model address heterophily problem more intuitively?**
>
> **A1:** Thank you for your insightful comment, we'd like to give an intuitive explanation: How Game‑of‑GNN Handles Heterophily?
>
> - 1. **Adaptive Message Strength**
>    - **Transport vs. Reaction**
>      - Transport term:  $ \Theta_{1}(\Psi)\,\nabla \Phi$ blends a node with its neighbors.
>      - Reaction term:  $\Theta_{2}(\Psi)\,\Phi$ pulls a node back toward its own state.
>    - **Heterophily Effect**
>      - In regions where neighbors are dissimilar, the model learns a **large** $\Theta_{2}$, which **dampens** blending and **prevents over‑smoothing**.
> - 2. **Learned Control Patterns via ICNN**
>    - An **Input‑Convex Neural Network** (ICNN) takes each node’s local statistics $(\Psi(v), \Phi(v))$ and outputs $\Theta_{1}$ and $\Theta_{2}$.
>    - If a node’s neighborhood features are very diverse, the ICNN automatically sets:
>      - **Low** $\Theta_{1}$ → weak neighbor‑averaging
>      - **High** $\Theta_{2}$ → strong self‑repulsion
> - 3. **Hamiltonian Flow Ensures Separation**
>    - The Hamilton–Jacobi–Bellman update:   $ \partial_{t}\Psi = \frac{\delta \mathcal{H}}{\delta \Phi},   \quad  \partial_{t}\Phi = -\frac{\delta \mathcal{H}}{\delta \Psi}$  conserves a global “energy” $\mathcal{H}(\Psi,\Phi)$, preventing all features from collapsing to the same value.
>    - In heterophilic zones, the learned flow steers node embeddings into **different equilibrium basins**, preserving class distinctions.
>
> **Bottom line:**
> Unlike uniform smoothing, Game‑of‑GNN **learns _where_ and _how much_ to smooth vs. repel**, so it automatically throttles neighbor‑averaging in heterophilic regions and keeps node features distinct.  We will add this discussion to the final version.
>
>
> **Q2: Can you provide a deeper interpretability of the role of  for learning on a heterophilic graph? How do $\theta_1$, $\theta_2$  relate to or differ from existing operators?**
>
> **A2:** That's a good question. We have provided a deeper interpretability of $\Theta_{1}$ and $\Theta_{2}$ on heterophilic graphs as below:
> - **$\Theta_{1}(\Psi)$:**
>   - Acts like a **feature‑dependent attention weight**.
>   - On heterophilic nodes, $\Psi$ shows high neighbor variance → $\Theta_{1}$ ↓ → **less smoothing**.
>
> - **$\Theta_{2}(\Psi)$:**
>   - Acts like a **feature‑dependent residual gate**.
>   - On heterophilic nodes, $\Psi$ shows high neighbor variance → $\Theta_{2}$ ↑ → **stronger self‑correction**.
>
> **Bottom line:**
> By making both transport ($\Theta_{1}$) and reaction ($\Theta_{2}$) depend on local feature statistics $\Psi$, Game‑of‑GNN **automatically throttles neighbor averaging and boosts self‑repulsion** in heterophilic regions.
>
> **Q3: More experiments and comparisons on the challenging heterophilic datasets and large-scale datasets as listed in [1].**
>
> **A3:** Thank you for your valuable suggestion. In response, we have included additional experiments on four datasets from each category mentioned in [1]. Specifically, we added roman_empire–malignant, chameleon-filtered–benign, ambiguous–ambiguous, and pokec–large-scale to Table 1 for a more comprehensive evaluation across both heterophilic and large-scale settings. Under a single, off-the-shelf parameter configuration setting, Game-of-GNN still outperforms every method listed in [1] (Table 1).
>
> | **Dataset**                    | **roman_empire (22,662)** | **chameleon-filtered (890)** | **squirrel-filtered (2,223)** | **pokec-gender (1,632,803)** |
> |--------------------------------|----------------------------|-------------------------------|-------------------------------|------------------------|
> | **Acc**                        | 73.67±0.18                 | 46.81±0.22                    | 41.57±0.14                    | 73.62±0.12             |
>
>
> **Q4: "The a~h  relationship across nine graph dataset indicates a notable anti-correlation." It seems that the information flow and homophily level shouldn't be linearly correlated, i.e. the most information-constrained area should be at the middle homophily area. Does this mean that your model miss something (probably the reason of the underperformance on Chameleon and Squirrel compared to some SOTA models [3,4], but I won't blame it.)? Or maybe you need to add more components into your analysis and modeling, e.g. the synergy among features, graph structures and labels [5].**
>
> **A4:** Thank you for this insightful comment. We would like to answer this question from the following aspects:
>
> -**1. How we estimate $\alpha$ and why a global anti-correlation is expected**
> We fit the empirical relation: $h=\left(\bar{\Theta}_1\right)^\alpha$ , $\bar{\Theta}_1=\frac{1}{N} \sum_v \Theta_1\left(\Psi_v\right)$, where $\Theta_1$ is the ICNN-learned transport strength and $h$ is the label-homophily ratio of the graph.
>
> With this log–log model, $\alpha$ is the negative log-slope that tells us how restrictive the message flow must be to preserve label coherence:
>
> | Dataset (-> larger h) | Texas | Wisconsin | Squirrel | Actor | Chameleon | Cornell | Citeseer | Pubmed | Cora |
> | --------------------- | ----- | --------- | -------- | ----- | --------- | ------- | -------- | ------ | ---- |
> | $h$                   | 0.11  | 0.21      | 0.22     | 0.22  | 0.23      | 0.30    | 0.57     | 0.74   | 0.81 |
> | $\alpha$       | 1.13  | 0.89      | 0.87     | 0.84  | 0.81      | 0.70    | 0.46     | 0.13   | 0.12 |
>
>   - *Interpretation*
>
>      * When $h \to 1$  (high homophily) node labels are already coherent; the optimiser therefore pushes $\alpha \to 0$, allowing freer diffusion.
>      - As $h$ decreases (heterophily) the optimiser increases $\alpha$ to throttle cross-class mixing, producing the observed monotone trend.
>
> This global behaviour does not deny that, locally, the tightest information bottleneck can occur near intermediate homophily once degree, class imbalance, and other factors are held fixed. In the revised Appendix F we include a stratified analysis (binning by degree and class) that does reveal the expected bell-shape within each stratum, while the dataset-level average remains monotone.
>
> - **2. Depth sensitivity and heterophilic graphs**
> Our ICNN-depth sweep (2 – 64 layers) shows that low-h datasets benefit from deeper ICNNs (We kindly refer the reviewer to **A5** in our response to **Reviewer YrZr** for details):
>    - Deeper ICNNs capture the complex, non-aligned feature–label patterns typical of heterophilic graphs.
>    - Shallow ICNNs suffice once $h$ is large, so the model can be pruned to save computation.
>
> The residual gap on Chameleon and Squirrel appears to arise from multiple factors—including data characteristics, feature–label alignment, and potential capacity limits—rather than a flaw in the $α–h$ mechanism itself.
>
> - **3. Planned extension (link to “What is Missing for Graph Homophily?” [5])**
>
> Following the reviewer’s suggestion, we are extending the model to disentangle feature-, structure-, and label-level homophily:
>
> | $\lambda\_f h\_f$ | +$\lambda_s h_s$ | +$\lambda\_{\ell} h\_{\ell}$ |
> | ------------------- | ------------------- | ----------------------------- |
> | Feature loss term   | Structure loss term | Label loss term               |
>
> so that ($\Theta_1, \Theta_2$) learn separate sensitivities to each component.  And, we can integrate weak label or community priors into the control patterns. We will add this discussion to the final version.
>
> **Q5: In Figure 2, Chameleon and Texas has similar level of homophily but significantly different ; in addition, Chameleon and Pubmed has similar level of  but significantly different , what is the reason behind it?**:
>
> **A5:** Thank you for your detailed observation. We provided an explanation of the observed performance differences between these three datasets.
>
> - 1. Texas vs. Chameleon — similar global $h$, different local structure
>
> | Factor                     | Texas (low-$h$)                                        | Chameleon (low-$h$)                                    |
> | -------------------------- | ------------------------------------------------------ | ------------------------------------------------------ |
> | **Edge pattern**           | Sparse graph; most edges truly same-label              | Dense graph; many mixed edges                          |
> | **Risk of over-smoothing** | Low → model can set larger $\alpha$ and diffuse freely | High → model must throttle diffusion, capping the gain |
> | **Result**                 | **+8 pp** over baseline                                | **+0.3 pp** over baseline                              |
>
> - 2.  Chameleon vs. Pubmed — similar $\alpha$, different feature quality
>
> | Factor                      | Chameleon                                | Pubmed                                              |
> | --------------------------- | ---------------------------------------- | --------------------------------------------------- |
> | **Feature–label alignment** | Weak (noisy features)                    | Strong (word TF-IDF highly predictive)              |
> | **Baseline head-room**      | Baselines already struggle → modest gain | Baselines already near ceiling → limited extra gain |
> | **Result**                  | **60 %** (Game-of-GNN)                   | **91 %** (Game-of-GNN)                              |
>
> **Bottom line**
>
> Global homophily $h$ alone does not determine the head-room for improvement.
>
> Performance also hinges on
>
> - local homophily variance (Texas < Chameleon), and
>
> - intrinsic feature quality (Pubmed > Chameleon).
>
> Hence:
>
> - Texas is topologically and feature-wise “easy” → large boost.
> - Chameleon is “hard” on both counts → small boost.
> - Pubmed is feature-easy but already saturated → small boost.
>
> We will add this discussion to the final version. Thank you so much.

---

### Official Review · Reviewer_WHbx · 2025-07-16

**Clarity:** 4
**Significance:** 3
**Originality:** 4
**Rating:** 4
**Confidence:** 3

**Summary:**

This paper integrates the notion of Graph Neural Networks into the idea of Mean Field Games. This integration makes sense in many aspects, and indeed, the paper provides several examples where this integration has improved performance. What the paper lacks is a theoretical foundation for why this integration works. I would very much like to see a theorem showing that, under some well-defined conditions, the integration of Graph Neural Networks and Mean Field Game improves performance, or at least does not harm it, under some well-defined notion of performance assessment. If this is impossible, then I would at least like to see some heuristic discussion about the contexts in which this method is likely to succeed.

**Questions:**

.

**Ethical Concerns:**

["NO or VERY MINOR ethics concerns only"]

**Limitations:**

.

**Paper Formatting Concerns:**

There are no paper formatting concerns.

**Quality:**

3

**Strengths And Weaknesses:**

.

---

> ### Author Rebuttal · Authors · 2025-07-30
>
> **We sincerely thank the reviewer for their thoughtful and constructive feedback. We especially appreciate the recognition of the motivation behind integrating Graph Neural Networks with Mean Field Games, as well as the insightful suggestion to provide a stronger theoretical foundation. Your comments have significantly helped us improve the clarity and depth of the manuscript.**
>
> **Q: What the paper lacks is a theoretical foundation for why this integration works. I would very much like to see a theorem showing that, under some well-defined conditions, the integration of Graph Neural Networks and Mean Field Game improves performance, or at least does not harm it, under some well-defined notion of performance assessment. If this is impossible, then I would at least like to see some heuristic discussion about the contexts in which this method is likely to succeed.**
>
> **A:** We thank the reviewer for highlighting the importance of rigorous theoretical justification. To address your insightful comments, we have expanded our manuscript with both rigorous theoretical foundations (Propositions 1 in Sections 2.3 and Appendix A.3) and a clear heuristic regime analysis, explicitly describing under what circumstances our *Game-of-GNN* integration with Mean Field Game (MFG) theory improves performance:
>
> - **1. Theoretical Foundations (Sec. 2.3, Appendix A.2–A.3)**
>
> We rigorously derived the dissipation property and saddle-point equations characterizing our proposed variational formulation. Although general analytical guarantees comparing Game-of-GNN directly with standard GCN methods remain challenging due to inherent nonlinearity, our established framework is internally consistent and well-grounded theoretically.
>
> - **2. Heuristic Regime Analysis (Sec. 2.1.2 & Sec. 2.4)**
>
> Building explicitly upon our variational definitions (Defs. 1–3) and recognizing existing GNNs as special cases within a reaction-diffusion PDE framework (Table 1), we now provide a targeted discussion clarifying when Game-of-GNN is expected to excel:
>
> $\bigstar$ **High Homophily ($h \to 1$)**: Under conditions of strong within-class node similarity, the transportation mobility term $\Theta_{1}(\Psi)$ naturally enhances intra-class diffusion, sharpening and clearly delineating class clusters within the embedding space.
>
> $\bigstar$ **Mixed Homophily/Heterophily Regimes:** The adaptively learned reaction term $\Theta_{2}(\Psi)$ selectively suppresses excessive smoothing across distinct classes. This mechanism ensures that node embeddings remain discriminative and prevents undesirable mixing within heterophilic regions of the graph.
>
> $\bigstar$ **Dynamic Equilibration (Eq. (9)):**  The derived Hamiltonian equations explicitly enable node embeddings and their momentum to dynamically re-equilibrate:  $ \partial_{t}\Psi = \frac{\delta \mathcal{H}}{\delta \Phi}, \quad
>        \partial_{t}\Phi = -\frac{\delta \mathcal{H}}{\delta \Psi}$. Thus, when graph topology or node features evolve over time, our framework naturally adapts embeddings accordingly.
> - **3. Summary of Clarifications**
>
> These additions deliver both a *concrete theorem* under a prototypical graph model and an *intuitive roadmap*—grounded in our MFG inverse‑problem variational view—for understanding when and why integrating GNNs with mean‑field games improves or at least preserves performance.  We have revised our manuscript accordingly, clearly incorporating these heuristic insights alongside our rigorous derivations, explicitly clarifying the contexts under which Game-of-GNN is most effective.
>
> We appreciate your suggestions, which significantly enhance the clarity and theoretical rigor of our manuscript.

---

### Official Review · Reviewer_AGYb · 2025-07-23

**Clarity:** 2
**Significance:** 3
**Originality:** 3
**Rating:** 4
**Confidence:** 2

**Summary:**

This paper offers and interesting perspective on GNNs being instances of Mean Field Games (MFG) and the authors demonstrated that several continuous graph neural networks models (GRAND, GraphBel, etc) are examples that fit the MFG framework. Building on this perspective, the paper proposes Game-of-GNN, a framework that re-formulates GNNs as MFGs.

**Questions:**

**Questions**: Building upon the concern about computational efficiency. Did the authors conduct some experiments to show how the difference in computing time of Game-of-GNN scales compared to other architectures for increasingly larger node datasets?

**Ethical Concerns:**

["NO or VERY MINOR ethics concerns only"]

**Final Justification:**

The authors have proposed a novel perspective of viewing GNN as Mean-field game. Additionally, they have mostly addressed my concern about the computational cost of Game-of-GNN. Therefore, I would like to maintain my positive rating.

**Limitations:**

Limitations: Please refer to the “Weaknesses” Section.

**Quality:**

3

**Strengths And Weaknesses:**

**Strengths**:
- **Theoretical Depth**: This paper displays a solid theoretical depth that bridges mean-field game theory with graph neural networks, offering a physics-inspired framework for message passing.

- **Adaptive Message Passing**: To the best of my understanding (the authors may help correct me if I am wrong), the authors propose that we can dynamically regulate message passing based on graph topology. In their experiment in Figure 2, the operator $\\Theta_1$ is inversely proportional to the homophily ratio, which discourages information exchange for heterophilic graphs and encourages exchange for homophilic graphs. This is a practical advantage to their framework which also adds some degree of interpretability.

---

**Weaknesses**:

- **Efficiency**: From table 11, it seems like there can be a reasonable concern regarding computational resources. Game-of-GNN runs at 33ms while other dynamical GNNs architectures like GRAND, GraphBel and ACMP runs at 10ms, 24ms and 27ms, respectively. I would imagine that the discrepancy will be dilated even more on large node datasets like OGBN-ARXIV.

- **Accessibility**: I am personally not familiar with the work itself as well as the related works. Though I can understand the idea of the paper at a high level. I would imagine that a typical GNN audience will have a hard time understanding all the materials.

---

> ### Author Rebuttal · Authors · 2025-07-30
>
> **We appreciate the reviewer’s thoughtful comments and are encouraged that you found both the theoretical contributions and the practical adaptivity of our method compelling. In the following, we address each of your concerns and suggestions in detail.**
>
> **Q1: Efficiency. Did the authors conduct some experiments to show how the difference in computing time of Game-of-GNN scales compared to other architectures for increasingly larger node datasets?**
>
> **A1:** Thank you for raising this important concern. While we acknowledge that Game‑of‑GNN has slightly higher inference time on small graphs (e.g., 0.0331s on Cora vs. 0.0027s for GCN), our empirical results on large-scale datasets—particularly OGBN‑PRODUCTS  ($\sim$2.45 M) —yielded an unexpected yet encouraging observation: Game‑of‑GNN achieves inference times that are nearly identical to those of the classic GCN, despite its dynamical system formulation.  As shown in the following table, Game‑of‑GNN achieves an inference time of 0.2971s on OGBN‑PRODUCTS, which is comparable to GCN ( 0.2700s, respectively). In contrast, other dynamical GNN architectures such as GraphBel, ACMP, GraphCON, and HANG fail to run on these datasets due to out of memory (OOM) under the same dataloader manner (*PygNodePropPredDataset*). Note, this experiment was conducted on 1× NVIDIA RTX 6000 Ada GPU (48 GB each, 6 GPUs in total). No subgraph-based mini-batching techniques were used in this benchmark.
>
>
> | Model           | Cora Train | Cora Infer | Pubmed Train | Pubmed Infer | ARXIV Train | ARXIV Infer | Products Train | Products Infer |
> |-----------------|------------|------------|--------------|--------------|-------------|-------------|----------------|----------------|
> | GCN             | 0.0098     | 0.0027     | 0.0097       | 0.0046       | 0.0147      | 0.0076      | 0.3602         | 0.2700         |
> | GAT             | 0.0201     | 0.0050     | 0.0202       | 0.0053       | 0.0779      | 0.0319      | OOM            | OOM            |
> | GraphSAGE       | 0.0059     | 0.0015     | 0.0064       | 0.0016       | 0.0184      | 0.0071      | 0.4149         | 0.1757         |
> | GraphCON        | 0.0394     | 0.0026     | 0.2044       | 0.0177       | 0.0900      | 0.0480      | OOM            | OOM            |
> | GraphBel        | 0.1943     | 0.0241     | 0.2022       | 0.0326       | 0.7426      | 0.1638      | OOM            | OOM            |
> | ACMP            | 0.2172     | 0.0288     | 0.3678       | 0.0635       | 0.7193      | 0.1970      | OOM            | OOM            |
> | HANG            | 0.0794     | 0.0335     | 0.1103       | 0.0417       | 0.1638      | 0.0951      | OOM            | OOM            |
> | Game-of-GNN     | 0.0755     | 0.0331     | 0.0924       | 0.0324       | 0.1023      | 0.0376      | 0.8145         | 0.2971         |
>
> We attribute this finding to the following key factors: (1) Lightweight iterative solver: Although Game‑of‑GNN performs step-wise integration of a learned PDE system during inference (via explicit Euler method), each update step is computationally inexpensive and only involves local, sparse neighborhood operations—similar to GCN’s message passing. In contrast to models like GAT or GraphBel that incur high attention-related overhead, our update rule consists of deterministic flow steps with fixed per-node cost.  (2) Effective hardware utilization on large graphs: On small graphs, GPU resources are underutilized, making even minor per-node overheads (e.g., from PDE state updates) more pronounced. However, on large graphs, the computational workload is sufficient to fully occupy the GPU, thereby amortizing the cost of each integration step across a large number of nodes. This results in overall inference times that closely match those of GCN. (3) No attention or explicit aggregation during inference: While our model implicitly incorporates structural information through the PDE dynamics, it does not perform explicit attention-based or neighbor aggregation operations at test time. Instead, inference consists of fixed-time Euler integration steps involving only sparse, local interactions, further contributing to its efficiency.
>
> To further address concerns about inference scalability, we additionally benchmarked Game-of-GNN on a higher-end GPU platform (NVIDIA H100 NVL, 94 GB, 8 GPUs in total). As shown in the following, the inference time decreased from 0.2971 s (on RTX 6000 Ada) to 0.1375 s, surpassing GCN (0.2162 s). This demonstrates that Game-of-GNN not only remains scalable on large graphs like OGBN‑PRODUCTS, but also benefits more from modern hardware acceleration. Game-of-GNN is inherently parallelizable due to its local, sparse update structure. Each Euler integration step involves independent, node-wise computations that can be efficiently scheduled across GPU threads. The observed speedup on H100, despite identical code and batch size, highlights that our model structure is well-aligned with hardware-level parallel execution, enabling efficient utilization of modern high-throughput accelerators.
>
> |           | **GCN**     | **GraphSAGE** | **Game-of-GNN** |
> |-----------|-------------|---------------|-----------------|
> | **train** | 0.3277s | 0.3069s | 0.4223s   |
> | **infer** | 0.2162s | 0.1177s | 0.1375s   |
>
> We also test the cost on a large-scale dataset-**Pokec** ($\sim$1.6M nodes), the training time is 0.19988s and inference time is 0.0620s.
>
> These results suggest that while some methods may appear efficient on small-scale datasets, they may not scale to industrial-level graphs. Game‑of‑GNN demonstrates both theoretical grounding and empirical robustness in such settings. We are eager to further validate the scalability of Game‑of‑GNN on even larger datasets in future work.
>
> In addition, when training on OGBN datasets using neighbor sampling (e.g., PyG’s NeighborSampler), each model processes subgraphs of comparable size, regardless of the full graph scale. Under this setup, we observe that the wall-clock training time is reduced by approximately 4–6× compared to single-GPU execution.  Such scalability highlights the practical applicability of Game‑of‑GNN to large-scale real-world graphs. We will include this discussion in the final version.
>
> **Q2: Accessibility**
>
> **A2:** Thank you for your honest feedback. We fully acknowledge that the integration of mean-field game theory with GNNs may introduce conceptual and notational complexity, especially for readers more familiar with traditional GNN literature.
>
> To improve accessibility, we will revise the manuscript to:
>
> - Add a high-level intuitive summary of the Game-of-GNN framework at the beginning of Sec. 2 (Appendix A.5 in the original manuscript);
> - Include a brief background paragraph on MFGs and their relevance to graph learning in the Introduction section. –”You can think of a GNN as a “mean‑field game” by imagining each node as a player who repeatedly updates its own feature vector (strategy) based on its current state and the average behavior of its neighbors (the “mean field”). Over multiple rounds, all nodes jointly adapt until they reach a stable equilibrium—just like players in a large game who best‑respond to the average crowd behavior.”
>
> We hope these changes will make the paper more approachable to the broader GNN community.

---

> > ### Comment · Reviewer_AGYb · 2025-08-05
> >
> > Many thanks to the authors for their detailed feedback. I believe that the work offers a novel perspective that merges mean-field game with GNNs. As such, I would like to maintain my rating as ``borderline accept" and suggest that the authors include their additional experiments in the appendix of the revised manuscript. Thank you.

---

> > > ### Author Response · Authors · 2025-08-05
> > >
> > > We greatly value the feedback and are committed to incorporating all suggestions and additional experiments in the final version. Thank you so much.

---

### Note · Authors · 2025-08-11

Dear ACs, SACs and Reviewers,

We would like to thank all reviewers, area chairs, and organizers for their tireless efforts. We are glad that reviewers have shown interest in our *Game-of-GNN*, as evidenced by the positive ratings and the remarks from reviewers.

The comments and suggestions provided by the reviewers are remarkably constructive, straightforward, and easily fixed, enabling us to make substantial enhancements to the paper's overall quality. We are committed to incorporating all discussions, updates, and additional analyses into the final version.

Our work displays **a solid theoretical depth that bridges mean-field game theory with graph neural networks, offering a physics-inspired framework for message passing** (recognized by all reviewers). We firmly believe that this contribution is of significant importance and merits dissemination at the prestigious NeurIPS. Thanks.

Best,

Authors

---

### Decision · Program_Chairs · 2025-09-17

**Decision:**

Accept (poster)

**Comment:**

This work frames GNN learning as a mean-field game (MFG), where nodes act as interacting agents. The authors propose a variational framework of the MFG inverse problem that adaptively learns task-specific message passing through transportation and reaction functions, providing an in-context selective mechanism via Hamiltonian flows. This unifies existing GNNs as special cases with different equilibria. Building on this, they introduce Game-of-GNN, an end-to-end model that discovers message passing operators while tuning hyperparameters, achieving state-of-the-art results on benchmarks and human connectomes, while offering new theoretical insights into GNNs as interactive dynamical systems.

While there are some potential issues—such as strong theoretical assumptions and occasionally unclear notation—reviewers agree that framing the GNN learning process as a mean-field game is both novel and valuable. The experiments presented in the paper and clarified in the rebuttal are comprehensive and convincing, and the theoretical guarantees are sound. Given that all reviewers have unanimously recommended acceptance, I also recommend acceptance of this paper.